# LDARNet: DNA Adaptive Representation Network with Learnable Tokenization for Genomic Modeling

## Abstract

Genomic foundation models increasingly adopt large language model architectures, yet almost all rely on fixed tokenization schemes such as $k$-mers or BPE. These approaches impose arbitrary sequence boundaries and risk discarding biologically relevant signals. Recent work introduced dynamic hierarchical tokenization in an autoregressive setup, demonstrating the feasibility of adaptive tokenization but leaving masked language modeling and downstream evaluation unexplored. We present **LDARNet**, a 120M-parameter hierarchical genomic foundation model that adapts hierarchical compression to the masked language modeling paradigm. LDARNet combines BiMamba-2 state-space layers with selective attention and uses ratio-based regularization to learn stable token boundaries without supervised segmentation. We evaluate LDARNet through comprehensive fine-tuning across 27 diverse tasks from the Genomics Benchmarks and Nucleotide Transformer suites, comparing against state-of-the-art models spanning 8M-2.5B parameters. LDARNet achieves 11 of 18 wins among compact models (<300M parameters) – a 5.5-fold improvement over the next-best alternatives – and establishes overall best performance on 5 challenging histone modification tasks, surpassing even 2.5B-parameter competitors. Notably, LDARNet wins 7 of 10 histone modification benchmarks, demonstrating that learnable compression boundaries effectively capture the long-range dependencies critical for epigenetic regulation modeling. These findings provide evidence that adaptive tokenization under masked language modeling yields biologically meaningful representations, and highlight hierarchical compression as a promising direction for efficient and scalable genomic foundation models.

## 1 Introduction

The success of large language models (LLMs) has motivated the development of foundation models for genomics, where large-scale pretraining can transfer across diverse predictive tasks. Recent models such as DNABERT and Nucleotide Transformer (Ji et al., 2021; Lopez et al., 2023) demonstrate that pretrained encoders can generalize effectively to promoters, enhancers, and splice sites. However, most approaches rely on *fixed tokenization*, such as $k$-mers or byte-pair encoding (BPE). While these schemes are effective for text, they impose arbitrary boundaries on genomic data and lack clear biological grounding, raising the question of whether adaptive tokenization can capture functional signals more faithfully.

A recent line of work has begun to address this. H-Net (Hwang et al., 2025b) introduced dynamic hierarchical tokenization in an AR setup, showing that adaptive segmentation of the genome is feasible at scale. While impactful, H-Net focused primarily on demonstrating the modeling principle and did not systematically evaluate downstream biological utility. In particular, it remains unknown whether adaptive tokenization yields embeddings that are competitive with established genomic foundation models under MLM paradigm.

We address this gap with **LDARNet** (Learnable DNA Adaptive Representation Network), a hierarchical model that adapts the H-Net architecture to MLM pretraining. Our main contributions are:

- We adapt the H-Net dynamic tokenization architecture (Hwang et al., 2025b) from autoregressive generation to masked language modeling. Full code and model weights will be released upon publication.

- We demonstrate that hierarchical compression with dynamic boundary prediction enables compact models (120M parameters) to match or surpass models 10-20-fold larger, achieving best overall performance on 5 histone modification tasks against 2.5B-parameter competitors through comprehensive fine-tuning evaluation across 27 diverse genomic tasks.

- We establish that architectural generality – through learnable multi-scale compression – outperforms domain-specific optimization for general-purpose foundation models, with LDARNet achieving 11/18 wins on cross-species tasks versus 1-2 wins for human-genome-specialized alternatives of comparable scale.

## 2 RELATED WORKS

### 2.1 TECHNICAL FOUNDATIONS

A central challenge for foundation models in genomics and other non-linguistic domains lies in tokenization. Transformers (Vaswani et al., 2017) achieved remarkable success in NLP by operating over subword vocabularies, but this design presumes the existence of semantically meaningful and human-interpretable units such as words. For DNA and raw byte sequences, where no such segmentation exists, tokenization remains an open problem: fixed schemes such as $k$-mers introduce arbitrary boundaries, while byte-level encodings dramatically inflate sequence length.

Several works attempted to bypass tokenization through isotropic byte-level modeling. MambaByte (Wang et al., 2024) applied Mamba-2 layers directly to characters, while LlamaByte extended Transformers to raw sequences. Although these approaches eliminate external preprocessing, flat byte-level models typically underperform tokenized counterparts of comparable scale, suggesting that meaningful intermediate units are still needed. SpaceByte (Slagle, 2024) partially addressed this by introducing hand-crafted boundary heuristics (e.g., space delimiters) to form chunks, but such strategies remain domain-specific and inflexible.

The Hierarchical Network (H-Net) framework (Hwang et al., 2025b) reframed tokenization as a learnable problem. H-Net introduced dynamic chunking, jointly optimizing boundary detection and representation learning in a multi-stage hierarchy. This design replaced the traditional tokenization–LM–detokenization pipeline with an end-to-end architecture, demonstrating that adaptive chunking can outperform tokenized Transformers at comparable scale and improve data efficiency in settings with weak or arbitrary tokenization heuristics.

These developments establish the technical foundations for moving beyond fixed or handcrafted vocabularies. They highlight tokenization not as a preprocessing choice but as a central modeling challenge, motivating architectures that can learn biologically meaningful units directly from raw sequences.

### 2.2 DNA FOUNDATION MODELS

Large-scale self-supervised pretraining has been rapidly adopted in genomics, giving rise to a family of DNA foundation models. Early contributions such as DNABERT (Ji et al., 2021) demonstrated the utility of BERT-style masked language modeling on genomic data using fixed $k$-mer tokenization, establishing a strong baseline for sequence-based prediction tasks. Subsequent works such as the Nucleotide Transformer (NT) (Lopez et al., 2023) and its successor NTv2 (Dalla-Torre et al., 2025) scaled Transformer encoders from hundreds of millions to billions of parameters trained across multi-species genomes, demonstrating strong transferability of genomic embeddings but facing the quadratic context-length bottleneck inherent to attention. To mitigate this limitation, several works integrated more efficient sequence architectures. GENA-LM (Fishman et al., 2025) employed sparse attention to extend receptive fields, while Caduceus (Schiff et al., 2024) introduced BiMamba blocks with shared weights, leveraging state-space recurrence for efficient long-context modeling. HyenaDNA (Nguyen et al., 2023) proposed implicit long convolutions that support substantially longer contexts, and JanusDNA (Duan et al., 2025) combined AR efficiency with the bidirectionality of masked modeling in a hybrid Mamba–Attention Mixture-of-Experts design, enabling pre-

training on million-base sequences. Together, these architectures illustrate the trade-off gap between capacity, efficiency, and context length that continues to shape genomic foundation model design.

## 2.3 TOKENIZATION IN GENOMIC MODELS

Tokenization remains a critical yet unresolved challenge in genomic modeling, as DNA lacks the natural segmentation cues of language. Fixed $k$-mer approaches, exemplified by DNABERT (Ji et al., 2021) and the Nucleotide Transformer (Lopez et al., 2023), provided early baselines but rely on arbitrary and biologically unmotivated boundaries. Byte-level models such as HyenaDNA (Nguyen et al., 2023), Caduceus (Schiff et al., 2024), and JanusDNA (Duan et al., 2025) preserve nucleotide-level fidelity but suffer from excessive sequence length, high computational cost, and limited ability to capture higher-order motifs. Subword strategies using BPE, as in DNABERT-2 (Zhou et al., 2023) and GENA-LM (Fishman et al., 2025), introduce flexible variable-length units yet generate vocabularies that reflect statistical co-occurrence rather than biological semantics, limiting interpretability. More recent approaches aim for adaptive tokenization: MxDNA (Qiao et al., 2024) learns discontinuous and overlapping units through a mixture-of-experts convolutional design, while VQDNA (Li et al., 2024) employs vector quantization to induce hierarchical vocabularies that capture genomic motifs at multiple scales. While these methods demonstrate improved adaptability and often strong performance, open challenges remain: MxDNA involves more complex training dynamics, and VQDNA introduces additional computational requirements. Collectively, existing methods highlight both the centrality of tokenization in genomic foundation models and the lack of a principled, biologically grounded solution, motivating further exploration of adaptive strategies.

## 3 LDARNET ARCHITECTURE

We introduce LDARNet, a hierarchical foundation model for genomic sequences that extends the H-Net design (Hwang et al., 2025b) with several architectural innovations. While H-Net was originally developed for AR language modeling, our modifications adapt the framework to MLM and introduce bidirectional mechanisms that better align with the bidirectional nature of DNA.

At a high level, LDARNet retains the hierarchical encoder–main–decoder organization of H-Net but incorporates four major changes: (i) Mamba layers are replaced with *BiMamba-2* blocks with shared weights same as Caduceus model, (ii) attention mechanisms are non-causal, (iii) encoder/decoder stacks use BiMamba-2 while the main backbone uses Transformer layers, and (iv) both router and dechunking modules are extended to bidirectional variants. This design preserves H-Net's efficiency while improving both expressivity and stability for genomic modeling.

### 3.1 OVERVIEW

Like H-Net, LDARNet consists of stacked stages of encoders, a central backbone, and decoders, as figure 1 illustrates. Each stage compresses the sequence through a content-aware *chunking* operation, processes it at a reduced resolution, and restores fine-grained information through *dechunking*.

For an $S$-stage hierarchy, we denote encoders and decoders by $\mathcal{E}^s$ and $\mathcal{D}^s$ ($1 \leq s \leq S$), and the central backbone by $\mathcal{M}$. The overall forward process is:

$$\hat{x}^{s+1} = \mathcal{E}^s(x^s), \qquad \hat{z}^S = \mathcal{M}(x^S), \qquad \hat{z}^{s-1} = \mathcal{D}^s(z^s) \tag{1}$$

where compression and decompression are performed by Chunk and Dechunk modules. Unlike H-Net, all layers in LDARNet are trained with *non-causal masking*, enabling bidirectional context modeling required by the MLM objective.

### 3.2 SEQUENCE PROCESSING BLOCKS

#### 3.2.1 ENCODER AND DECODER BLOCKS: BIMAMBA-2

We extend the H-Net backbone by replacing its causal Mamba layers with a *bidirectional*, non-causal variant of Mamba-2, which we term **BiMamba-2**. This design preserves the linear-time recurrence of state-space models while enabling full-context conditioning, which is essential for MLM and other non-autoregressive objectives.

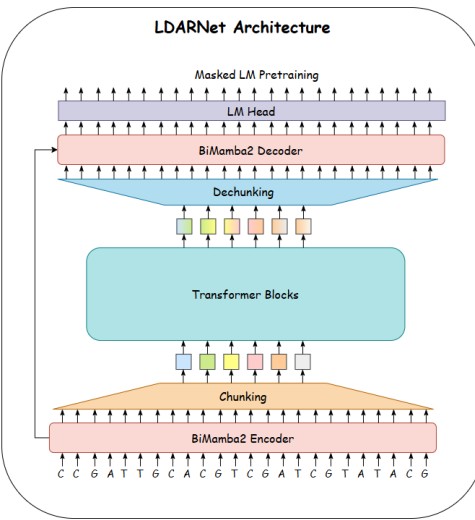 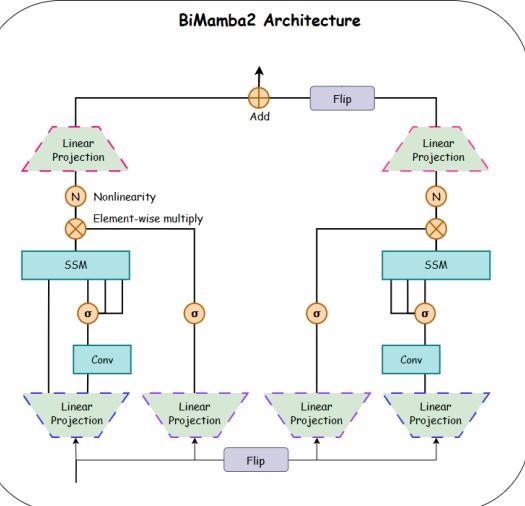

Figure 1: **Model overview.** Left: the LDARNet architecture with BiMamba outer layers and a Transformer backbone operating in a compressed latent space. Right: the internal structure of a BiMamba-2 block used in the outer networks.

**Mamba-2 as selective state-space layers.** Mamba-2 (Dao & Gu, 2024) instantiates a *selective* state-space layer whose dynamics are conditioned on the input. The model admits both a linear recurrent formulation and a quadratic dual representation via structured semiseparable (SS) matrices, a property referred to as SSD duality. For input $x_t \in \mathbb{R}^D$ and hidden state $h_t \in \mathbb{R}^N$:

$$h_{t+1} = \bar{A}_t h_t + \bar{B}_t x_t, \qquad y_t = C_t h_t + D x_t \tag{2}$$

$$\bar{A}_t, \ \bar{B}_t = \text{discretize}(A, B_t, \Delta_t), \tag{3}$$

$$B_t = W_B x_t, \quad C_t = W_C x_t, \quad \Delta_t = \text{softplus}(W_\Delta x_t) \tag{4}$$

Efficient GPU kernels implement block-SS decompositions and fused projections, supporting large $N$ with stable training and favorable wall-clock efficiency.

**Bidirectional construction with mean fusion.** To enable bidirectional context aggregation, we apply a Mamba-2 (M2) cell both in the forward and reversed temporal order, fusing the outputs by mean-pooling. Given input $X \in \mathbb{R}^{B \times T \times D}$ and padding mask $M \in \{0, 1\}^{B \times T}$:

$$Y = \tfrac{1}{2}\Big[ \text{M2}(X \odot M) \ + \ \text{flip}_T\big(\text{M2}(\text{flip}_T(X \odot M))\big) \Big] \tag{5}$$

Here $\odot$ applies masking across features and $\text{flip}_T$ denotes temporal reversal. The mean fusion avoids introducing additional parameters while preserving symmetry across directions.

**Parameter tying.** A naive bidirectional construction would double the number of parameters, since separate forward and reverse Mamba-2 modules would each maintain their own projections. However, most parameters reside in the input and output projection layers rather than in the convolution or SSM submodules (Gu & Dao, 2024). To avoid this overhead, BiMamba-2 shares these dominant weights across directions:

$$\{W_{\text{in}}, b_{\text{in}}, W_{\text{out}}, b_{\text{out}}\}^{\rightarrow} \ = \ \{W_{\text{in}}, b_{\text{in}}, W_{\text{out}}, b_{\text{out}}\}^{\leftarrow}. \tag{6}$$

This weight tying ensures that the forward and reverse passes instantiate a single shared Mamba-2 definition, yielding a parameter-efficient block that also respects reverse-complement symmetry in genomic sequences.

**Comparison to Transformers.** Whereas Transformers rely on quadratic attention to obtain bidirectional context, BiMamba-2 attains the same global conditioning in *linear time* via structured recurrence. This yields a more scalable encoder block, retaining expressivity while offering efficiency gains for long-context genomic modeling.

### 3.2.2 MAIN BACKBONE.

The core backbone $\mathcal{M}$ operates over compressed representations ($L^S \ll L^0$), where tokens encode higher-level semantic abstractions. We instantiate $\mathcal{M}$ with Transformer layers for two reasons: (i) self-attention provides an effective mechanism for modeling long-range dependencies among compressed tokens, and (ii) it enables direct comparability with established BPE-based Transformer baselines in genomics (Ji et al., 2021; Lopez et al., 2023). This hybrid design – BiMamba modules in the outer stages and Transformers at the core – yields a principled balance between efficiency and expressivity, combining the scalability of state-space models with the representational flexibility of attention.

## 3.3 DYNAMIC CHUNKING AND DECHUNKING

A central innovation of our architecture lies in adapting the dynamic chunking mechanism of H-Nets (Hwang et al., 2025b) to a *bidirectional MLM setting*. We introduce two key modifications: (i) the *router* employs bidirectional similarity to detect boundaries symmetrically, and (ii) the *dechunker* incorporates a bidirectional exponential moving average (EMA) smoother for more stable reconstruction.

**Bidirectional Routing.** Given encoder outputs $\hat{x}_t \in \mathbb{R}^D$, the router projects them into query–key pairs,

$$q_t = W_q \hat{x}_t, \quad k_t = W_k \hat{x}_t, \tag{7}$$

with $W_q, W_k \in \mathbb{R}^{D \times D}$ initialized as identity matrices to ensure numerical stability at early training. Unlike the unidirectional formulation in H-Net, we compute both forward and backward cosine similarities:

$$s_t^{\text{fwd}} = \cos(q_t, k_{t+1}), \quad s_t^{\text{bwd}} = \cos(q_t, k_{t-1}), \tag{8}$$

which are then averaged to produce a symmetric similarity score. Boundary probabilities are defined as

$$p_t = \tfrac{1}{2}\left(1 - \tfrac{1}{2}(s_t^{\text{fwd}} + s_t^{\text{bwd}})\right), \tag{9}$$

with $p_t \in [0, 1]$. High discontinuity between neighbors (low cosine similarity) yields a strong boundary signal. From these probabilities we derive a hard boundary mask $b_t = 1_{\{p_t \geq 0.5\}}$. Padding positions are forced to $p_t = 1.0$ to ensure proper chunk alignment. This router produces both soft probabilities (for differentiable training) and hard masks (for inference).

**Chunking.** The chunking operator compresses the sequence by retaining only tokens marked as boundaries:

$$x^{s+1} = \{\hat{x}_t \mid b_t = 1\}, \quad p^{s+1} = \{p_t \mid b_t = 1\}. \tag{10}$$

This implements a hierarchical downsampling mechanism where chunk selection is data-dependent rather than fixed.

**Bidirectional Dechunking with EMA.** Reconstruction from compressed tokens is inherently unstable due to discrete boundary decisions. We therefore extend H-Net's dechunker with a *bidirectional EMA smoother*.

Let $z_j$ denote compressed representations and $p_j$ their boundary probabilities. We compute decay factors as

$$\Delta_j = -\log(1 - p_j), \tag{11}$$

and propagate states with EMA dynamics:

$$\bar{z}_t^{\text{fwd}} = p_t z_t + (1 - p_t)\,\bar{z}_{t-1}^{\text{fwd}}, \quad \bar{z}_t^{\text{bwd}} = p_t z_t + (1 - p_t)\,\bar{z}_{t+1}^{\text{bwd}}. \tag{12}$$

The final reconstruction averages forward and backward passes,

$$\bar{z}_t = \tfrac{1}{2}(\bar{z}_t^{\text{fwd}} + \bar{z}_t^{\text{bwd}}), \tag{13}$$

ensuring symmetry and robustness. For efficient sequence propagation, we leverage the fused selective scan kernel (`mamba_chunk_scan_combined`).

**Upsampling.** To restore the sequence to its original length $L^s$, we assign each position $t$ the representation of its corresponding compressed token. Specifically, let $j(t) = \sum_{k=1}^{t} b_k$ denote the cumulative boundary count up to position $t$, which serves as the index into the compressed sequence. The upsampled representation is then:

$$\tilde{z}_t = \bar{z}_{j(t)}, \quad \text{where } j(t) = \sum_{k=1}^{t} b_k. \tag{14}$$

This operation broadcasts each compressed token across its corresponding span in the original sequence, providing a differentiable approximation of discrete boundary expansion.

### 3.4 TRAINING OBJECTIVE

Unlike H-Net's AR training, LDARNet is optimized under a MLM loss, which is more appropriate for bidirectional DNA modeling. The overall objective is:

$$\mathcal{L} = \mathcal{L}_{\text{MLM}} + \alpha \sum_{s=0}^{S-1} \mathcal{L}_{\text{ratio}}^s, \tag{15}$$

where the first term is the standard cross-entropy loss for MLM, and the second term regularizes the compression ratio at each stage to avoid degenerate chunking solutions.

**Ratio Loss.** We adopt the ratio loss from H-Net (Hwang et al., 2025b), originally introduced to prevent trivial compression behavior:

$$\mathcal{L}_{\text{ratio}} = \frac{N}{N-1} \left( (N-1)FG + (1-F)(1-G) \right), \quad F = \frac{1}{L} \sum_{t=1}^{L} b_t, \quad G = \frac{1}{L} \sum_{t=1}^{L} p_t, \tag{16}$$

where $F$ is the fraction of vectors actually selected, $G$ is the average boundary probability, and $N$ is the target compression ratio. By construction, the minimum of $\mathcal{L}_{\text{ratio}}$ occurs when $F = G = 1/N$, yielding $\mathcal{L}_{\text{ratio}} = 1$. However, as noted in Hwang et al. (2025b), the loss can in principle fall below 1 when $F \neq G$ (e.g. $F = 1/N + \epsilon$, $G = 1/N - \epsilon$), which we also observe empirically.

In practice, this regularizer effectively guides the model toward balanced compression while preserving adaptivity to biologically meaningful regions. Combined with the MLM loss, it enables LDARNet to learn non-trivial, context-dependent segmentations rather than collapsing to fixed heuristics.

### 3.5 MODEL IMPLEMENTATION AND PRETRAINING

We instantiate **LDARNet** as a 120M-parameter single-stage hierarchical model with compression ratio $N = 4$. The architecture follows an encoder-main backbone-decoder structure [m3t1, [M10], m4]: three BiMamba-2 layers and one local attention layer in the encoder, ten BiMamba-2 layers in the main backbone, and four BiMamba-2 layers in the decoder. Model dimensions scale across the hierarchy with $d_{\text{model}} = 512$ in outer stages and $d_{\text{model}} = 768$ in the main backbone, following the principle that compressed representations benefit from greater channel capacity. The vocabulary comprises seven byte-level tokens: $\{A, C, G, T, N, \texttt{[PAD]}, \texttt{[MASK]}\}$.

**Training.** We employ masked language modeling (MLM) with 15% masking probability, combining reconstruction loss with a ratio-based regularizer ($\alpha = 0.03$) that encourages the boundary predictor to maintain the target compression ratio. Models are optimized using AdamW Loshchilov & Hutter (2017) with base learning rate $5 \times 10^{-4}$ and a warmup-stable-decay (WSD) schedule: 10% warmup, 70% plateau, 20% decay. Following Hwang et al. (2025b), we apply stage-wise learning rate scaling to outer layers ($3\times$ multiplier) to compensate for gradient attenuation through compression boundaries. Training uses effective batch size 512 on sequences of length 4096.

**Corpus.** The pretraining data combines the human reference genome with the multispecies collection from Nucleotide Transformer (Dalla-Torre et al., 2025), ensuring both in-species fidelity and cross-species diversity. Each sequence is sampled in forward and reverse orientations with equal probability to promote reverse–complement invariance.

Complete training details are provided in Appendix A, with ablation studies in Appendix B.

| Task | Enformer 252M | DNABERT-2 117M | HyenaDNA 55M | Caduceus-Ph 8M | Caduceus-PS 8M | GROVER 87M | LDarNet 120M | NT-multi 2.5B | NT-v2 500M | Generator 1.2B | Generator-All 1.2B |
|---|---|---|---|---|---|---|---|---|---|---|---|
| *# Wins* | 0 | 2 | | 2 | | | 11 | - | - | - | - |
| H3 | 72.4 ± 1.8 | 78.5 ± 1.2 | 78.1 ± 1.5 | $\underline{79.4 \pm 1.2}$ | 77.2 ± 2.2 | 76.8 ± 0.8 | 78.2 ± 1.2 | 79.3 ± 1.3 | 78.8 ± 1.0 | **80.6 ± 0.5** | 80.3 ± 0.7 |
| H3K14ac | 28.4 ± 2.4 | 51.5 ± 0.9 | **$\underline{60.8 \pm 2.0}$** | 56.4 ± 3.3 | 59.6 ± 3.8 | 54.8 ± 2.0 | 58.9 ± 3.6 | 53.8 ± 0.9 | 53.8 ± 1.5 | 60.5 ± 0.8 | 58.0 ± 3.8 |
| H3K36me3 | 34.5 ± 1.9 | 59.1 ± 0.5 | 61.4 ± 1.4 | 59.0 ± 1.8 | 61.1 ± 4.8 | 56.3 ± 1.7 | $\underline{62.4 \pm 0.7}$ | 61.8 ± 1.1 | 61.8 ± 1.5 | **65.7 ± 0.7** | 63.1 ± 1.3 |
| H3K4me1 | 29.1 ± 1.6 | 51.2 ± 0.8 | 51.2 ± 0.8 | 46.8 ± 1.5 | 48.7 ± 2.9 | 46.1 ± 1.8 | $\underline{58.3 \pm 1.2}$ | 54.1 ± 0.5 | 54.4 ± 0.9 | 55.3 ± 0.9 | 54.9 ± 1.8 |
| H3K4me2 | 20.7 ± 2.1 | 33.3 ± 1.3 | 45.5 ± 2.8 | 33.2 ± 3.4 | 43.1 ± 1.6 | 40.3 ± 4.2 | $\underline{49.6 \pm 1.4}$ | 32.4 ± 1.4 | 30.2 ± 2.0 | 42.4 ± 1.3 | 40.0 ± 1.5 |
| H3K4me3 | 15.6 ± 2.2 | 35.3 ± 2.1 | 55.0 ± 1.5 | 49.0 ± 4.2 | 52.8 ± 3.3 | 45.8 ± 2.2 | $\underline{57.6 \pm 4.3}$ | 40.8 ± 1.1 | 43.7 ± 2.8 | 51.2 ± 0.9 | 47.3 ± 4.7 |
| H3K79me3 | 49.8 ± 1.3 | 61.5 ± 1.0 | 66.9 ± 1.4 | 64.1 ± 2.8 | 68.2 ± 1.8 | 62.6 ± 2.6 | **$\underline{68.7 \pm 2.5}$** | 62.3 ± 1.0 | 62.1 ± 1.2 | 67.0 ± 1.1 | 63.1 ± 2.1 |
| H3K9ac | 41.5 ± 2.0 | 54.5 ± 0.9 | 58.6 ± 2.1 | 57.5 ± 2.4 | 56.4 ± 1.8 | 58.1 ± 1.5 | $\underline{60.3 \pm 2.1}$ | 54.7 ± 1.1 | 56.7 ± 2.0 | **61.2 ± 0.6** | 60.3 ± 1.9 |
| H4 | 73.5 ± 2.3 | 79.7 ± 0.8 | 76.3 ± 1.2 | 78.8 ± 1.0 | 79.9 ± 1.0 | 76.9 ± 1.7 | $\underline{81.3 \pm 1.1}$ | 80.8 ± 0.7 | 79.5 ± 0.8 | **81.5 ± 0.8** | 80.8 ± 1.0 |
| H4ac | 27.5 ± 2.2 | 46.5 ± 1.3 | 56.4 ± 1.1 | 54.8 ± 2.7 | 58.5 ± 1.8 | 53.0 ± 1.7 | **$\underline{62.3 \pm 1.4}$** | 49.2 ± 1.4 | 50.2 ± 2.5 | 59.2 ± 1.5 | 56.5 ± 3.5 |
| Enhancer | 45.4 ± 2.9 | 52.5 ± 2.6 | 52.0 ± 3.1 | 52.2 ± 2.4 | 51.1 ± 2.6 | 51.6 ± 1.8 | $\underline{57.7 \pm 1.4}$ | 54.5 ± 2.8 | 56.1 ± 2.9 | **58.0 ± 1.5** | 54.0 ± 2.6 |
| Enhancer type | 31.2 ± 4.3 | 42.3 ± 1.8 | 40.3 ± 5.6 | 40.3 ± 2.8 | 41.0 ± 2.6 | $\underline{43.3 \pm 2.9}$ | 42.0 ± 2.7 | 44.4 ± 2.2 | 44.4 ± 3.6 | **47.7 ± 1.7** | 46.3 ± 2.3 |
| Promoter all | 91.0 ± 0.4 | $\underline{94.5 \pm 0.3}$ | 91.9 ± 0.3 | 93.7 ± 0.2 | 94.1 ± 0.3 | 93.9 ± 0.7 | 93.2 ± 0.5 | 95.1 ± 0.4 | 95.2 ± 0.2 | **96.2 ± 0.2** | 95.5 ± 0.2 |
| Promoter non-TATA | 91.0 ± 0.6 | $\underline{94.4 \pm 0.3}$ | 91.9 ± 0.4 | 93.5 ± 0.7 | 94.0 ± 0.2 | 92.5 ± 0.6 | 94.4 ± 0.5 | 95.5 ± 0.3 | 95.2 ± 0.3 | **96.2 ± 0.1** | 95.5 ± 0.2 |
| Promoter TATA | 92.0 ± 1.2 | 91.1 ± 1.1 | 88.1 ± 2.0 | 89.5 ± 1.0 | 90.3 ± 1.0 | 89.1 ± 0.9 | $\underline{92.3 \pm 0.5}$ | 91.9 ± 0.8 | 93.3 ± 0.9 | **94.8 ± 0.8** | 93.1 ± 0.7 |
| Splice acceptor | 77.2 ± 0.7 | 90.9 ± 0.4 | $\underline{93.5 \pm 0.5}$ | 91.8 ± 1.7 | 90.7 ± 1.5 | 92.7 ± 0.9 | 91.2 ± 1.0 | 97.3 ± 0.2 | 97.3 ± 0.4 | **98.1 ± 0.2** | 95.7 ± 0.9 |
| Splice site all | 83.1 ± 1.2 | 95.0 ± 0.3 | 91.7 ± 0.6 | 93.5 ± 1.1 | $\underline{95.3 \pm 0.5}$ | 91.9 ± 0.5 | 94.2 ± 1.6 | 97.4 ± 0.4 | 97.5 ± 0.2 | **97.8 ± 0.1** | 97.3 ± 0.2 |
| Splice donor | 81.3 ± 1.5 | 92.7 ± 0.3 | 89.4 ± 1.3 | 91.2 ± 0.9 | $\underline{93.0 \pm 1.0}$ | 88.8 ± 1.2 | 92.8 ± 1.9 | 97.4 ± 0.2 | 97.7 ± 0.7 | **97.8 ± 0.2** | 96.7 ± 0.5 |

Table 1: **Nucleotide Transformer tasks comparison.** Models are grouped by size: < 300M parameters (left) and ≥ 300M parameters (right). **Bold** indicates the best result overall, underlined indicates the best result among models < 300M. Best performing model < 300M: LDARNet (11/18 wins). Values shown as mean ± std across folds.

| Benchmark | DNABERT-2 117M | HyenaDNA 55M | Caduceus-Ph 8M | Caduceus-PS 8M | GROVER 87M | LDarNet 120M | NT-v2 500M | Generator 1.2B | Generator-All 1.2B |
|---|---|---|---|---|---|---|---|---|---|
| *# Wins* | 3 | 0 | 2 | 2 | 0 | 3 | - | - | - |
| Coding vs. Intergenomic | 95.1 ± 0.2 | 90.2 ± 0.4 | 93.3 ± 0.1 | 94.4 ± 0.2 | 91.9 ± 0.2 | $\underline{95.5 \pm 0.1}$ | 95.5 ± 0.1 | **96.3 ± 0.0** | 95.9 ± 0.1 |
| Drosophila Enhancers Stark | 77.4 ± 1.1 | 77.0 ± 1.6 | **$\underline{82.7 \pm 1.0}$** | 81.6 ± 1.5 | 76.1 ± 1.1 | 81.0 ± 0.8 | 79.7 ± 0.9 | 82.1 ± 0.5 | 76.8 ± 1.5 |
| Human Enhancers Cohn | $\underline{75.8 \pm 0.5}$ | 72.5 ± 0.9 | 74.7 ± 0.3 | 74.9 ± 0.3 | 73.8 ± 0.3 | 75.2 ± 0.3 | 75.6 ± 0.6 | **76.3 ± 0.2** | 75.4 ± 0.6 |
| Human Enhancers Ensembl | 91.8 ± 0.3 | 90.1 ± 0.3 | **$\underline{92.4 \pm 0.2}$** | 92.3 ± 0.2 | 91.1 ± 0.4 | 90.6 ± 0.7 | 92.1 ± 0.4 | 91.7 ± 0.2 | 91.2 ± 0.2 |
| Human Ensembl Regulatory | 87.4 ± 0.7 | 93.2 ± 0.1 | 93.8 ± 0.4 | $\underline{94.1 \pm 0.2}$ | 89.7 ± 0.1 | **$\underline{94.1 \pm 0.1}$** | 94.1 ± 0.1 | 92.8 ± 0.1 | 92.6 ± 0.1 |
| Human non-TATA Promoters | 95.7 ± 0.8 | 89.4 ± 2.3 | 96.1 ± 0.3 | 96.1 ± 0.2 | 95.0 ± 0.5 | **$\underline{96.3 \pm 0.4}$** | 93.2 ± 0.6 | 95.8 ± 0.1 | 95.5 ± 0.5 |
| Human OCR Ensembl | 80.6 ± 0.3 | 77.4 ± 0.4 | 82.5 ± 0.4 | $\underline{82.6 \pm 0.3}$ | 78.9 ± 0.2 | 79.8 ± 0.3 | 81.3 ± 0.1 | 82.3 ± 0.2 | 81.2 ± 0.3 |
| Human vs. Worm | $\underline{97.7 \pm 0.1}$ | 95.8 ± 0.4 | 97.5 ± 0.1 | 97.6 ± 0.1 | 96.6 ± 0.1 | 97.6 ± 0.0 | 97.6 ± 0.1 | **98.0 ± 0.0** | 97.8 ± 0.1 |
| Mouse Enhancers Ensembl | $\underline{86.5 \pm 1.4}$ | 75.6 ± 3.0 | 78.8 ± 2.8 | 82.6 ± 2.1 | 74.2 ± 2.5 | 78.2 ± 2.6 | 85.5 ± 1.8 | **87.1 ± 1.5** | 78.4 ± 2.7 |

Table 2: **Genomic Benchmarks comparison.** Models are grouped by size: <300M parameters (left) and ≥300M parameters (right). **Bold** indicates the best result overall, underlined indicates the best result among models <300M. Best performing model <300M: DNABERT-2 (3/9 wins). Values shown as mean ± std across folds.

## 3.6 Downstream Evaluation

To rigorously assess LDARNet's learned representations, we evaluate on two comprehensive benchmark suites: the **Nucleotide Transformer (NT) tasks** (Dalla-Torre et al., 2025) with 18 diverse datasets spanning histone modifications, regulatory elements, and splice sites, and **Genomic Benchmarks (GB)** (Grešová et al., 2023) with 9 classification tasks focused on regulatory genomics. These benchmarks probe a wide range of genomic functions across varying sequence lengths and biological contexts.

**Evaluation setup.** We adopt the rigorous experimental protocol from Generator (Wu et al., 2025), which provides the most comprehensive comparison framework to date. Specifically, we uniformly fine-tune all models with 10-fold cross-validation on all datasets. For each model-task pair, we conduct exhaustive hyperparameter search over 9 learning rates and 4 batch sizes (36 configurations total), select the best-performing configuration on validation data, then report test metrics from 10-fold cross-validation using this optimal configuration. This two-stage procedure ensures both optimal performance and statistical robustness. Details are in Appendix A.4.

**Models compared.** We benchmark LDARNet against state-of-the-art genomic foundation models, grouping them by scale: **compact models** (<300M parameters) include Enformer (252M) (Avsec et al., 2021), DNABERT-2 (117M) (Zhou et al., 2023), HyenaDNA (55M) (Nguyen et al., 2023), Caduceus-Ph and Caduceus-PS (8M each) (Schiff et al., 2024), GROVER (87M) (Sanabria et al., 2024), and LDARNet (120M); **large-scale models** (≥300M) include NT-multi (2.5B) and NT-v2 (500M) (Dalla-Torre et al., 2025), and Generator variants (1.2B) (Wu et al., 2025). Model details are in Appendix A.6.

**Results.** Table 1 shows NT task results. **Among compact models, LDARNet achieves state-of-the-art performance with 11 out of 18 wins** – a 5.5× improvement over the next-best compact alternatives (DNABERT-2, HyenaDNA, Caduceus-PS: 2 wins each). LDARNet's superiority is particularly pronounced on histone modification tasks, winning 7 out of 10 tasks. Remarkably, on five tasks (H3K4me1, H3K4me2, H3K4me3, H3K79me3, H4ac), **LDARNet achieves the best overall result, surpassing even models 20× larger**. This exceptional performance on chromatin-related tasks suggests that hierarchical compression effectively captures the long-range dependencies and multi-scale patterns critical for modeling epigenetic regulation.

Table 2 shows GB results. GB tasks exhibit high baseline performance (most models >90%, many >95%), making differentiation challenging. Nevertheless, **LDARNet ties with DNABERT-2 for best compact model performance (3/9 wins each)**. Notably, on Human non-TATA Promoters, LDARNet achieves 96.3% accuracy – the best overall result across all models, including those 10× larger. Caduceus models (8M) show surprisingly strong GB performance (2 wins each), attributable to their exclusive training on human genome. However, this specialization limits cross-species generalization, as evidenced by weaker NT performance.

**Summary.** Across 27 diverse genomic tasks, LDARNet establishes itself as the leading compact genomic foundation model with 11/18 NT wins and 3/9 GB wins (tied for best among compact models). Critically, LDARNet frequently achieves overall best results even against models with 10-20× more parameters, validating that hierarchical compression with dynamic width scheduling enables efficient modeling of multi-scale genomic patterns without sacrificing representational capacity. Detailed per-task analysis is provided in Appendix A.7.

## 4 RESULTS

**Nucleotide Transformer Tasks.** On the 18-task NT benchmark (Table 1), **LDARNet achieves 11 wins among compact models (<300M parameters) – representing a 5.5-fold improvement over the next-best alternatives**. Notably, LDARNet secures 7 of 10 histone modification tasks and establishes overall best performance on 5 tasks (H3K4me1, H3K4me2, H3K4me3, H3K79me3, H4ac), surpassing even 2.5B-parameter models. These results demonstrate that hierarchical compression with dynamic boundary selection effectively captures the long-range dependencies and multi-scale patterns essential for epigenetic regulation modeling. On regulatory element and splice site prediction, LDARNet attains the best compact-model performance on Enhancer classification (57.7 MCC) while maintaining competitive results on promoter identification.

**Genomic Benchmarks.** On the 9-task GB suite (Table 2), LDARNet achieves parity with DNABERT-2 for best compact model performance (3 wins each). GB tasks are characterized by high baseline accuracies (>90-95%), creating a ceiling effect that limits performance differentiation. Nevertheless, **on Human non-TATA Promoters, LDARNet achieves 96.3% accuracy – the highest result across all evaluated models**, exceeding Generator-1.2B (95.8%) and NT-v2-500M (93.2%). Caduceus models (8M parameters) exhibit competitive GB performance attributable to human-genome-specific training, yet this specialization constrains cross-species generalization, evidenced by only 1-2 NT wins compared to LDARNet's 11 – illustrating the fundamental trade-off between domain specialization and architectural generality.

**Parameter Efficiency.** With 120M parameters, LDARNet consistently matches or exceeds the performance of models containing 500M-2.5B parameters. On 5 NT tasks, it establishes best overall results despite 4-20-fold parameter disadvantages. These findings validate that strategic architectural innovations – hierarchical compression, dynamic width scheduling, and reversible embeddings – achieve performance parity with naive parameter scaling while requiring substantially reduced computational resources.

## 5 CONCLUSION

We present **LDARNet**, a hierarchical genomic foundation model integrating learnable sequence compression with dynamic width scheduling for efficient multi-scale sequence modeling. Compre-

hensive evaluation across 27 downstream tasks establishes LDARNet as the state-of-the-art compact model (<300M parameters), achieving 11/18 wins on NT tasks and 3/9 wins on GB tasks.

**Key contributions**: (1) Hierarchical compression with dynamic boundary prediction enables compact models to match or surpass models 10-20-fold larger, achieving best overall performance on 5 histone modification tasks against 2.5B-parameter competitors. (2) Adaptation of hierarchical network architecture (Hwang et al., 2025b) for masked language modeling, providing the community with an efficient alternative to autoregressive pretraining that will be released with full code upon publication. (3) Comprehensive evaluation across 27 diverse genomic tasks demonstrating that architectural generality – through learnable multi-scale compression – outperforms domain-specific optimization for general-purpose foundation models, with 11/18 cross-species wins versus 1-2 wins for human-specialized alternatives.

LDARNet's superior histone modification performance (7/10 wins) provides empirical evidence that learnable compression boundaries align with biologically meaningful sequence segmentation. The performance dichotomy between NT and GB benchmarks is informative: on multi-species NT tasks requiring cross-species generalization, LDARNet's architecture-driven generality dominates (11 wins); conversely, on human-centric GB tasks, genome-specialized models like Caduceus achieve competitive performance (2 wins each) at the cost of limited transferability (1-2 NT wins). For general-purpose genomic foundation models, architectural generality demonstrates greater value than task-specific optimization.

**Future directions**: Extension to multi-stage hierarchical compression would enable ultra-long genomic context processing (>100kb) while preserving computational tractability. Zero-shot and few-shot evaluation protocols would complement supervised fine-tuning assessments, providing deeper characterization of learned representation transferability. Multimodal integration with orthogonal genomic measurements (RNA-seq, ATAC-seq, Hi-C) could exploit hierarchical structure for cross-modal dependency modeling at multiple biological scales.

**Broader impact**: LDARNet establishes that compact, efficiently-designed models achieve performance parity with substantially larger alternatives, lowering barriers to genomic AI adoption and enabling deployment in resource-constrained and latency-sensitive applications. These results indicate that continued progress in genomic foundation models need not depend exclusively on parameter count escalation – principled architectural innovation offers a complementary and more sustainable development trajectory as genomic datasets scale and application diversity increases.

## 6 LIMITATIONS

While LDARNet establishes state-of-the-art performance among compact genomic foundation models, several limitations warrant discussion.

**Performance on specific task categories.** Large-scale models (Generator-1.2B, NT-v2-500M) maintain advantages on splice site prediction tasks, where local pattern memorization may be more critical than long-range dependency modeling. This suggests that certain genomic functions benefit more from parameter count than from architectural innovation, indicating potential limits to efficiency gains through compression alone.

**Single-stage compression.** Our implementation employs single-stage 4× compression. Multi-stage hierarchical compression could enable processing of ultra-long genomic contexts (¿100kb), which are essential for modeling large-scale chromatin interactions and structural variants. However, maintaining gradient flow and representational capacity across multiple compression stages requires careful architectural design that we leave to future work.

**Evaluation scope.** Our assessment focuses on supervised fine-tuning for classification tasks. Zero-shot and few-shot evaluation protocols would provide complementary insights into representation transferability and generalization. Additionally, we evaluated sequences up to 4096bp; systematic evaluation on longer contexts would better characterize the efficiency advantages of hierarchical compression at scale.

**Interpretability of learned boundaries.** While LDARNet's strong histone modification performance suggests that learned compression boundaries align with biologically meaningful segmentation, we have not systematically analyzed these boundaries. Correlating learned token bound-

aries with experimentally validated genomic features (e.g., transcription factor binding sites, chromatin accessibility peaks) would provide deeper mechanistic insights into what the model learns and strengthen biological interpretability.

## IMPACT STATEMENT

This work demonstrates that strategically designed compact models can achieve performance parity with substantially larger alternatives in genomic sequence modeling. By establishing that 120M-parameter models can match or exceed 2.5B-parameter models on challenging tasks, we lower barriers to genomic AI adoption, enabling researchers with limited computational resources to deploy competitive foundation models. The 10-20× reduction in computational requirements has practical implications for resource-constrained environments and real-time applications, from clinical diagnostics to high-throughput functional genomics. We will release full model weights and training code upon publication.

However, as with all advances in genomic AI, our work carries potential for dual use. While LDAR-Net can accelerate beneficial applications in precision medicine and biological discovery, the same capabilities could theoretically be misapplied for designing harmful biological sequences. We emphasize the importance of responsible development practices, including appropriate access controls and ongoing dialogue between the ML and biosecurity communities to mitigate potential risks while preserving the substantial benefits of genomic AI for human health and scientific understanding.

## LLM USAGE

LLMs were employed to assist with the preparation of this manuscript. Specifically, we used OpenAI's GPT models to improve the clarity, coherence, and grammar of the text, and to help rephrase sections for consistency with academic writing standards. All technical content, experimental design, data analysis, and results interpretation were conceived, implemented, and validated by the authors. The LLM was not used to generate novel scientific ideas, design experiments, or analyze results. Final responsibility for the accuracy and integrity of the content rests entirely with the authors.

## ETHICS STATEMENT

This work focuses on the development and evaluation of machine learning models for genomic sequence modeling. All datasets used are publicly available reference genomes or previously released benchmark collections; no private, identifiable, or clinical human data were used. Our methods are intended for basic research in machine learning and genomics, and do not directly provide medical diagnoses or clinical recommendations. We acknowledge that advances in genomic foundation models could have dual-use implications, including both positive applications (e.g., improved understanding of gene regulation, variant effect prediction) and potential risks if misapplied. To mitigate risks, we release models and code under a research license and encourage responsible use aligned with scientific and biomedical research goals.

## REPRODUCIBILITY STATEMENT

We provide comprehensive documentation to ensure full reproducibility of our results. All architectural specifications, including layer configurations, dimensional parameters, and attention settings, are detailed in Section 3.5 with complete hyperparameter listings in Appendix A. Training procedures, including optimizer configuration, learning rate schedules, stage-wise scaling factors, and infrastructure details (6× NVIDIA A100 80GB GPUs), are fully specified in Appendix A.

Downstream evaluation protocols, including the exhaustive 36-configuration hyperparameter search and 10-fold cross-validation procedure, are described in Section 3.6 with implementation details in Appendix A.4. Complete results are presented in Tables 1 and 2, with per-task analysis in Appendix A.7.

Upon publication, we will publicly release: (1) full model architecture and training code, (2) pre-trained model checkpoints, (3) fine-tuning scripts for all evaluated tasks, and (4) evaluation pipelines with exact hyperparameter configurations used for each model-task pair. These resources will enable researchers to replicate our results, compare against LDARNet on new benchmarks, and extend the architecture to additional genomic applications.

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

## A  TRAINING DETAILS

### A.1  ARCHITECTURE SPECIFICATION

LDARNet is instantiated with approximately 120M parameters following the hierarchical layout [m3t1, [M10], m4]. The encoder comprises three BiMamba-2 layers followed by one local attention layer, the main backbone contains ten BiMamba-2 layers, and the decoder consists of four BiMamba-2 layers. This asymmetric design prioritizes efficient sequence processing at full resolution while concentrating representational capacity in the compressed domain.

Model dimensions increase across the hierarchy to account for the reduced sequence length in compressed space. Outer stages (encoder and decoder) use $d_{\text{model}} = 512$, while the main backbone operates with $d_{\text{model}} = 768$. The feed-forward network in the backbone employs hidden dimension $d_{\text{ff}} = 2560$. BiMamba-2 state-space modules are configured with chunk size 256, convolutional kernel width 4, state dimension $d_{\text{state}} = 128$, and expansion factor 2.

Attention layers use 16 heads with rotary position embeddings of dimension 32 in outer stages and 48 in the main backbone. We employ local windowed attention with window size 1023 in outer stages to maintain computational efficiency, while the main backbone uses global attention to enable full-context reasoning over compressed representations. The vocabulary comprises seven byte-level tokens ($\{$A, C, G, T, N, [PAD], [MASK]$\}$) with untied input and output embeddings.

### A.2  TRAINING OBJECTIVE AND OPTIMIZATION

The pretraining objective combines masked language modeling with compression regularization:

$$\mathcal{L} = \mathcal{L}_{\text{MLM}} + \alpha \cdot \mathcal{L}_{\text{ratio}}, \tag{17}$$

where $\mathcal{L}_{\mathrm{MLM}}$ is the standard reconstruction loss over masked positions, and $\mathcal{L}_{\mathrm{ratio}}$ is a ratio-based loss that encourages the boundary predictor to select approximately $1/N$ positions for retention. We set $\alpha = 0.03$ to balance reconstruction quality with compression consistency. Tokens are masked with 15% probability during training.

Models are optimized using AdamW Loshchilov & Hutter (2017) with $\beta_1 = 0.9$, $\beta_2 = 0.95$, $\epsilon = 10^{-8}$, and weight decay 0.01. Gradients are clipped to maximum norm 1.0. We employ a warmup-stable-decay (WSD) learning rate schedule with base rate $5 \times 10^{-4}$: linear warmup over the first 10% of training, constant plateau for 70%, followed by inverse square-root decay over the final 20% to 5% of the peak value.

Following prior work on hierarchical models Hwang et al. (2025b), we apply stage-wise learning rate scaling to account for gradient magnitude differences across compression boundaries. Outer-stage parameters (encoder and decoder) receive scaled learning rate $\eta_{\mathrm{outer}} = \eta_{\mathrm{base}} \cdot \sqrt{N} \cdot (d_{\mathrm{back}}/d_{\mathrm{outer}})$, which compensates for gradient attenuation through the compressed backbone. For our configuration with $N = 4$, $d_{\mathrm{outer}} = 512$, and $d_{\mathrm{back}} = 768$, this yields a $3\times$ multiplier for outer layers relative to the backbone.

Training uses gradient accumulation over 16 steps with micro-batch size 32, yielding an effective batch size of 512 sequences per iteration. All sequences are of length 4096 tokens.

## A.3 TRAINING INFRASTRUCTURE

Training is performed using PyTorch DistributedDataParallel (DDP) with NCCL backend across multiple GPUs. We enable hardware-accelerated matrix operations (TF32 on Ampere-generation GPUs) and memory-efficient attention kernels (FlashAttention Dao et al. (2022)) when available. Mixed-precision training uses `bfloat16` on compatible hardware with automatic fallback to `float16` with gradient scaling on older architectures.

Input sequences are sampled from human genomic coordinates specified in BED format, with data distributed across workers using stratified sampling to ensure balanced epoch coverage. The pre-training corpus combines the human reference genome (GRCh38/hg38) with the multispecies collection from Nucleotide Transformer (Dalla-Torre et al., 2025), ensuring both in-species fidelity and cross-species diversity. To promote reverse–complement invariance, each sequence is sampled in forward and reverse orientations with equal probability.

## A.4 DOWNSTREAM EVALUATION SETUP

Following the comprehensive evaluation protocol established by Generator (Wu et al., 2025), we uniformly fine-tune all models using 10-fold cross-validation on all benchmark tasks. For each model on each dataset, we conduct an exhaustive hyperparameter search over learning rates in $\{1e^{-5}, 2e^{-5}, 5e^{-5}, 1e^{-4}, 2e^{-4}, 5e^{-4}, 1e^{-3}, 2e^{-3}, 5e^{-3}\}$ and batch sizes in $\{64, 128, 256, 512\}$. Early stopping is applied based on validation loss with patience of 5 epochs. This exhaustive search ensures that each model achieves its optimal performance on every task, making the comparison particularly fair.

For each task, we select the hyperparameter configuration that achieves the best validation performance, then report test metrics averaged over all cross-validation folds with standard deviation computed across folds. This protocol ensures statistical robustness and eliminates potential confounds from suboptimal hyperparameter selection.

## A.5 OPTIMAL HYPERPARAMETERS PER TASK

After exhaustive hyperparameter search over 36 configurations (9 learning rates × 4 batch sizes) per model-task pair, we identified the optimal configuration for LDARNet on each downstream task. Tables A.5 and A.5 report the learning rate and effective batch size that achieved best validation performance for each task. These configurations were then used for 10-fold cross-validation to produce the final test results reported in the main text.

**Task-specific observations.** Several patterns emerge from the optimal configurations. Histone modification tasks generally prefer moderate learning rates ($1e^{-3}$ to $2e^{-3}$) with smaller to medium

| Task | Learning Rate | Batch Size |
|------|---------------|------------|
| *Histone Modifications* | | |
| H3 | $1 \times 10^{-3}$ | 64 |
| H3K4me1 | $5 \times 10^{-3}$ | 128 |
| H3K4me2 | $5 \times 10^{-3}$ | 512 |
| H3K4me3 | $1 \times 10^{-3}$ | 64 |
| H3K9ac | $1 \times 10^{-3}$ | 128 |
| H3K14ac | $1 \times 10^{-3}$ | 128 |
| H3K36me3 | $5 \times 10^{-4}$ | 64 |
| H3K79me3 | $2 \times 10^{-3}$ | 128 |
| H4 | $2 \times 10^{-3}$ | 64 |
| H4ac | $1 \times 10^{-3}$ | 64 |
| *Regulatory Elements* | | |
| Enhancer | $2 \times 10^{-4}$ | 128 |
| Enhancer type | $2 \times 10^{-3}$ | 64 |
| Promoter all | $1 \times 10^{-3}$ | 128 |
| Promoter non-TATA | $5 \times 10^{-3}$ | 256 |
| Promoter TATA | $1 \times 10^{-3}$ | 128 |
| *Splice Sites* | | |
| Splice acceptor | $1 \times 10^{-3}$ | 128 |
| Splice donor | $2 \times 10^{-3}$ | 64 |
| Splice site all | $5 \times 10^{-3}$ | 128 |

Table 3: **Optimal hyperparameters for Nucleotide Transformer tasks.** Learning rate (LR) and effective batch size (BS) selected from exhaustive search over 36 configurations.

| Task | Learning Rate | Batch Size |
|------|---------------|------------|
| Coding vs. Intergenomic | $1 \times 10^{-3}$ | 64 |
| Drosophila Enhancers Stark | $2 \times 10^{-3}$ | 128 |
| Human Enhancers Cohn | $2 \times 10^{-3}$ | 64 |
| Human Enhancers Ensembl | $1 \times 10^{-3}$ | 128 |
| Human Ensembl Regulatory | $2 \times 10^{-3}$ | 64 |
| Human non-TATA Promoters | $1 \times 10^{-3}$ | 128 |
| Human OCR Ensembl | $2 \times 10^{-3}$ | 64 |
| Human vs. Worm | $1 \times 10^{-3}$ | 64 |
| Mouse Enhancers Ensembl | $1 \times 10^{-3}$ | 128 |

Table 4: **Optimal hyperparameters for Genomic Benchmarks tasks.** Learning rate (LR) and effective batch size (BS).

batch sizes (64-128), except H3K4me2 which benefits from larger batches (512). Regulatory element tasks show more diversity: enhancer classification requires conservative learning ($2e^{-4}$), while promoter tasks span the full range from $1e^{-3}$ to $5e^{-3}$. Splice site tasks consistently prefer aggressive learning rates ($1e^{-3}$ to $5e^{-3}$) with moderate to large batches (64-128), possibly reflecting their more localized signal patterns. GB tasks predominantly cluster around $1e^{-3}$ to $2e^{-3}$ with batch sizes 64-128, consistent with their focus on human regulatory elements with strong baseline performance.

## A.6 MODEL ARCHITECTURES AND TRAINING DETAILS

This section provides detailed specifications for all baseline models evaluated in our downstream benchmarks.

**Compact Models (< 300M parameters). Enformer (252M)** (Avsec et al., 2021) is a Transformer-based model originally trained in a supervised manner specifically for chromatin profile and gene expression prediction tasks. Unlike other baselines that use unsupervised pretraining, Enformer was directly optimized on ENCODE and Roadmap Epigenomics data. This task-specific training may explain its continued competitiveness on chromatin-related benchmarks despite its earlier release. The model employs standard absolute positional encodings and processes sequences up to 196kb with convolutional downsampling.

**DNABERT-2 (117M)** (Zhou et al., 2023) employs Byte-Pair Encoding (BPE) tokenization with a learned vocabulary, combined with ALiBi (Attention with Linear Biases) positional encoding for extrapolation to longer sequences. The model was pretrained on a multi-species genome corpus using masked language modeling. BPE tokenization allows the model to learn subword units that may capture biologically meaningful motifs, though at the cost of losing single-nucleotide granularity.

**HyenaDNA (55M)** (Nguyen et al., 2023) represents a departure from Transformer architectures, utilizing implicit long convolutions inspired by state-space models. The model employs single-nucleotide tokenization and was pretrained on the human reference genome. Its convolutional nature enables efficient processing of very long sequences (up to 1M bp during pretraining) with subquadratic complexity.

**Caduceus-Ph and Caduceus-PS (8M each)** (Schiff et al., 2024) are bidirectional Mamba models – the smallest models in our comparison by a substantial margin. Both variants employ single-nucleotide tokenization and were trained exclusively on the human reference genome (GRCh38). The "Ph" variant uses a phase-based bidirectional architecture, while "PS" uses a parallel scan approach. Their compact size and human-specific training make them particularly efficient for human genomic tasks, though potentially limiting cross-species generalization.

**GROVER (87M)** (Sanabria et al., 2024) combines BPE tokenization with specialized pretraining objectives beyond standard masked language modeling. The model incorporates domain-specific inductive biases for genomic sequences and was pretrained on a diverse genomic corpus.

**Large-Scale Models (≥300M parameters). NT-multi (2.5B) and NT-v2 (500M)** (Dalla-Torre et al., 2025) employ k-mer tokenization with masked language modeling. Both models were pretrained on a comprehensive multi-species genomic corpus. Notably, NT-v2, despite being 5× smaller than NT-multi, demonstrates enhanced performance on many benchmarks, suggesting that recent architectural improvements can substantially improve parameter efficiency.

**Generator (1.2B) and Generator-All (1.2B)** (Wu et al., 2025) are autoregressive models trained on comprehensive genomic data with next-token prediction. Generator was trained on a curated genomic corpus, while Generator-All incorporated additional data sources.

## A.7 DETAILED RESULTS ANALYSIS

**Nucleotide Transformer Tasks: Histone Modifications.** The NT benchmark includes 10 histone modification prediction tasks, which are particularly challenging due to the long-range dependencies involved in chromatin organization. LDARNet achieves remarkable performance on these tasks, winning 7 out of 10 benchmarks and achieving overall best results on 5 tasks.

On H3K4me1 (58.3), H3K4me2 (49.6), H3K4me3 (57.6), H3K79me3 (68.7), and H4ac (62.3), LDARNet not only leads among compact models but achieves the best overall result across all models, including those with 2.5B parameters. This is particularly noteworthy given that these models have 20× more parameters. The consistent excellence on H3K4 methylation marks (me1, me2, me3) suggests that LDARNet's hierarchical architecture is especially well-suited for capturing the multi-scale patterns associated with active promoter and enhancer regions.

On H3K9ac (60.3) and H4 (81.3), LDARNet achieves the best compact-model results and comes within 1-2 MCC points of the best large-scale models (Generator: 61.2 and 81.5, respectively). On H3K36me3, LDARNet (62.4) ranks as the best compact model, though Generator (65.7) achieves a more substantial lead. The only histone task where LDARNet does not lead among compact models is H3K14ac, where HyenaDNA achieves a remarkable 60.8 MCC – actually the best overall result across all models, demonstrating that even much smaller models can occasionally achieve breakthrough performance on specific tasks.

The strong performance on histone modification tasks validates our hypothesis that hierarchical compression with dynamic boundary selection enables effective modeling of long-range chromatin interactions. These tasks require understanding dependencies spanning thousands of base pairs, which conventional models struggle to capture efficiently.

**Nucleotide Transformer Tasks: Regulatory Elements and Splice Sites.** On regulatory element prediction (Enhancer, Enhancer type, Promoter variants), LDARNet demonstrates consistent competitiveness. It achieves the best compact-model result on Enhancer (57.7 MCC), closely approaching Generator (58.0). DNABERT-2 shows particular strength on promoter tasks, winning on Promoter all (94.5) and Promoter non-TATA (94.4), likely benefiting from its BPE tokenization which may better capture promoter-specific motifs.

Splice site prediction tasks (Splice acceptor, Splice donor, Splice site all) are dominated by large-scale models, with Generator achieving >97.5% on all three. Among compact models, Caduceus-PS shows strength on these tasks (winning Splice site all and Splice donor), possibly due to its bidirectional architecture being well-suited for the highly localized patterns around splice junctions. HyenaDNA achieves the best compact-model result on Splice acceptor (93.5), demonstrating its effectiveness on tasks requiring precise local pattern recognition.

**Genomic Benchmarks: Saturation Effects and Specialized Models.** GB tasks present a different challenge than NT tasks due to performance saturation. On 6 out of 9 tasks, the best-performing model achieves >92% accuracy, and on 4 tasks, even compact models exceed 95%. This narrow margin makes it difficult to demonstrate clear architectural advantages.

Nevertheless, LDARNet achieves competitive performance with 3 wins. On Coding vs. Intergenomic task (95.5%), LDARNet ties with NT-v2 for the best compact-model result, approaching Generator (96.3%). On Human Ensembl Regulatory task (94.1%), LDARNet achieves a three-way tie for the overall best result with Caduceus-PS and NT-v2, demonstrating that compact models can match large-scale performance in certain regimes. Most impressively, on Human non-TATA Promoters task (96.3%), LDARNet achieves the single best result across all models, outperforming Generator (95.8%) and NT-v2 (93.2%).

Caduceus models show surprisingly strong GB performance despite being 15× smaller than LDAR-Net, achieving 2 wins each and several overall best results (Drosophila Enhancers Stark task: 82.7%, Human Enhancers Ensembl task: 92.4%, Human OCR Ensembl task: 82.6%). This performance can be directly attributed to their exclusive training on the human genome – they are effectively specialist models for human genomic tasks. However, this specialization comes at the cost of generalization: on NT tasks, which include cross-species data, Caduceus models achieve only 1-2 wins each, substantially underperforming LDARNet's 11 wins.

DNABERT-2 demonstrates well-balanced performance across both benchmarks, achieving 2 NT wins and 3 GB wins. Its strength on enhancer and cross-species tasks (Human Enhancers Cohn, Human vs. Worm, Mouse Enhancers Ensembl) suggests that BPE tokenization with multi-species pretraining produces robust, generalizable representations.

**Performance-Parameter Efficiency Analysis.** A key finding from our evaluation is that LDAR-Net achieves exceptional parameter efficiency. Across both benchmarks, LDARNet (120M) frequently matches or exceeds models with 500M-2.5B parameters. On 5 NT tasks, LDARNet achieves the overall best result despite being 4-20× smaller than its closest competitors. This efficiency validates our architectural approach: rather than simply scaling up parameter count, strategic architectural innovations – hierarchical compression, dynamic width scheduling, reversible embeddings – can achieve comparable or superior performance at a fraction of the computational cost.

The contrast between Caduceus and LDARNet is particularly instructive. Caduceus models (8M) achieve strong GB performance through domain specialization (human-only training), while LDAR-Net (120M) achieves strong performance across both human-centric GB and multi-species NT through architectural generality. This suggests two distinct paths to efficiency: specialization (narrow but deep optimization for specific domains) versus generalization (broad competence through architectural innovation). For genomic foundation models intended for diverse downstream applications, the generalist approach appears more promising.

## A.8 COMPUTATIONAL BUDGET

**Pretraining.** LDARNet (120M parameters) was pretrained on 6× NVIDIA A100 80GB GPUs with mixed-precision training (`bfloat16`). The corpus combines human reference genome (GRCh38) and multispecies data from Nucleotide Transformer (Dalla-Torre et al., 2025) (∼300B base pairs). Training used sequences of length 4096, effective batch size 512, and required 7 days wall-clock time, totaling **1,008 GPU-hours** (42 GPU-days).

**Downstream Evaluation.** For each task, we performed exhaustive hyperparameter search over 36 configurations (9 learning rates × 4 batch sizes) followed by 10-fold cross-validation with the optimal configuration. Training time per configuration ranged from 30 minutes to 6 hours depending on task complexity and dataset size, averaging 2 hours.

- Hyperparameter search: 27 tasks × 36 configs × 2h = 1,944 GPU-hours (81 GPU-days)
- Cross-validation: 27 tasks × 10 folds × 2h = 540 GPU-hours (22.5 GPU-days)
- Downstream total: 2,484 GPU-hours (103.5 GPU-days)

**Total Cost.** Complete experimental pipeline required **3,492 GPU-hours (145.5 GPU-days)** on A100 80GB hardware. For comparison, pretraining a 2.5B-parameter model (NT-multi scale) requires approximately 1,000-1,500 GPU-days, demonstrating that LDARNet achieves competitive performance at <10% of large-scale model training costs. At 400W TDP per A100 with datacenter PUE 1.2, total energy consumption is approximately 1,680 kWh, corresponding to ∼670 kg COe at US grid average carbon intensity.

## B ABLATION STUDIES

To systematically evaluate design choices while conserving computational resources, all ablation experiments were conducted using 2M-parameter models with identical architectural configurations. Each model was trained for 10 epochs on the same data splits, enabling direct comparison across experimental conditions. We evaluate downstream performance using Matthews Correlation Coefficient (MCC) on two benchmark suites: Nucleotide Transformer (NT) tasks and Genomic Benchmarks (GB). For interpretability, we partition NT tasks into histone modification prediction (NT Histones) and regulatory element classification comprising enhancers, promoters, and splice sites (NT Regulatory).

**Training Setup.** All ablation models were trained on human genomic sequences using byte-level tokenization with MLM at 15% masking probability. We employed a composite loss function combining MLM loss with a ratio-based ratio loss weighted by $\alpha$. Models were optimized using AdamW ($\beta_1 = 0.9$, $\beta_2 = 0.95$, $\epsilon = 10^{-8}$) with a base learning rate of $5 \times 10^{-4}$ and weight decay of 0.01. We applied a warmup-stable-decay (WSD) learning rate schedule comprising 10% warmup, 70% stable training, and 20% decay phases, with the final learning rate reaching 5% of the peak value. Training utilized automatic mixed precision (bfloat16 on Ampere GPUs, float16 otherwise), gradient accumulation over 16 steps yielding an effective batch size of 256, and gradient clipping at maximum norm 1.0. Distributed training was performed using PyTorch DistributedDataParallel with NCCL backend, with TF32 operations and FlashAttention enabled for computational efficiency.

**Downstream Evaluation Setup.** To assess the quality of learned representations, we employ a linear probing protocol following established practices in genomic foundation model evaluation. For each pretrained model, we extract embeddings by performing a forward pass through the encoder and applying mean pooling over the sequence length dimension, weighted by the attention mask to exclude padding tokens. The resulting fixed-dimensional representations are then used to train a logistic regression classifier (L-BFGS solver, maximum 1000 iterations) without fine-tuning the encoder weights.

We evaluate on two benchmark suites: Nucleotide Transformer (NT) downstream tasks and Genomic Benchmarks (GB). For NT tasks, we employ 10-fold stratified cross-validation on the training set with evaluation on the held-out test split. For GB datasets, we use 5-fold stratified cross-

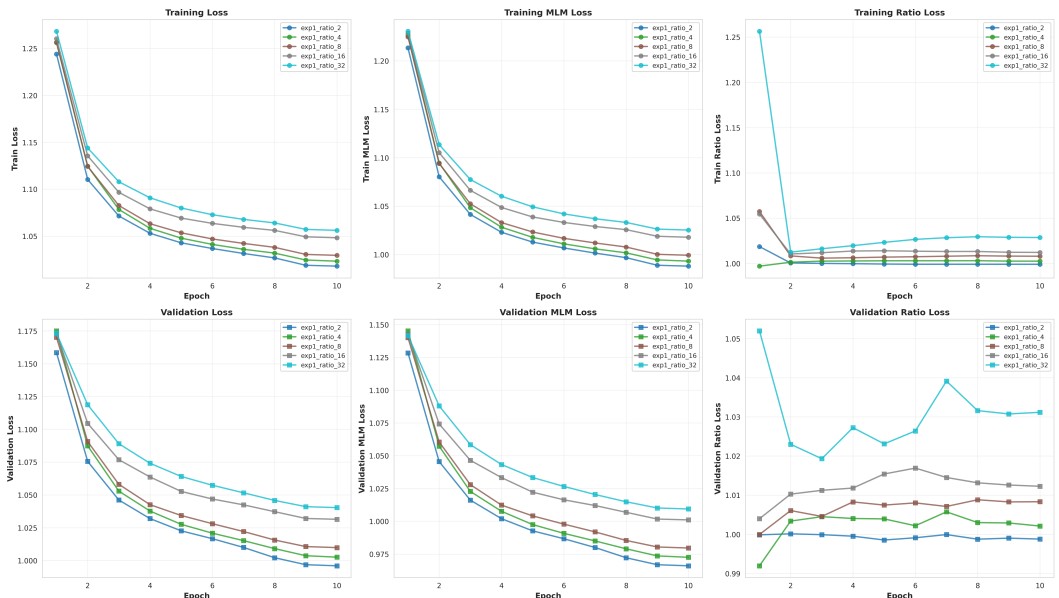

Figure 2: Training loss curves across compression ratios $N \in \{2, 4, 8, 16, 32\}$. Higher compression ratios consistently yield elevated loss values, reflecting the fundamental trade-off between compression efficiency and information preservation.

validation. We report Matthews Correlation Coefficient (MCC) as the primary metric, which provides a balanced measure for potentially imbalanced classification tasks. Confidence intervals (95%) are computed using the Student's t-distribution across folds. All experiments use identical hyperparameters and random seeds to ensure fair comparison.

## B.1 COMPRESSION RATIO

The compression ratio $N$ determines the degree of sequence compression, directly affecting the information throughput of the model. We evaluated compression ratios $N \in \{2, 4, 8, 16, 32\}$ to characterize the trade-off between compression efficiency and representational capacity.

Figure 2 illustrates the relationship between compression ratio and training loss. As expected, higher compression ratios yield increased loss values, reflecting the reduced information capacity of more aggressive compression. This behavior is consistent with rate-distortion theory: stronger compression necessarily discards information, manifesting as degraded reconstruction fidelity.

Downstream evaluation reveals a more nuanced pattern (Tables 5–7). While $N = 32$ achieves the highest average MCC on histone modification tasks (39.75%), $N = 2$ leads on NT Regulatory tasks (50.93%), and $N = 4$ achieves the best performance on Genomic Benchmarks (52.07%). This task-dependent variation suggests that optimal compression depends on the characteristic length scales of regulatory features: histone modifications may benefit from aggressive compression that emphasizes global patterns, while promoter and splice site recognition requires finer-grained local information.

We selected $N = 4$ for the main experiments as it achieves the best balance across benchmark suites, with competitive performance on all task categories (average ranks: 2.0 on NT Histones, 2.0 on NT Regulatory, 1.0 on GB) while maintaining moderate compression that preserves both local and long-range dependencies.

## B.2 RATIO LOSS WEIGHT

The ratio loss, weighted by $\alpha$, regularizes the learned compression toward the target ratio $N$. We investigated $\alpha \in \{0.0, 0.03, 0.1, 0.3\}$ at fixed $N = 4$, and additionally evaluated $\alpha \in \{0.0, 0.03\}$ at $N = 8$ to examine the interaction between ratio loss weight and compression ratio.

| Task | ratio_2 | ratio_4 | ratio_8 | ratio_16 | ratio_32 |
|---|---|---|---|---|---|
| *Average* | 39.16 | 39.24 | 39.41 | 38.22 | **39.75** |
| *# Wins* | 2 | 1 | 1 | 1 | **5** |
| **H3** | $66.48 \pm 0.20$ | $\mathbf{67.69 \pm 0.17}$ | $67.68 \pm 0.23$ | $66.36 \pm 0.28$ | $66.88 \pm 0.19$ |
| **H3K14ac** | $27.23 \pm 0.31$ | $27.03 \pm 0.25$ | $27.71 \pm 0.26$ | $25.16 \pm 0.22$ | $\mathbf{28.88 \pm 0.13}$ |
| **H3K36me3** | $38.04 \pm 0.17$ | $38.77 \pm 0.27$ | $39.31 \pm 0.21$ | $35.64 \pm 0.25$ | $\mathbf{39.81 \pm 0.26}$ |
| **H3K4me1** | $30.36 \pm 0.26$ | $29.56 \pm 0.20$ | $29.44 \pm 0.16$ | $\mathbf{32.00 \pm 0.23}$ | $30.10 \pm 0.21$ |
| **H3K4me2** | $\mathbf{26.69 \pm 0.20}$ | $24.23 \pm 0.22$ | $24.89 \pm 0.31$ | $24.53 \pm 0.22$ | $24.44 \pm 0.14$ |
| **H3K4me3** | $20.09 \pm 0.36$ | $19.37 \pm 0.26$ | $18.28 \pm 0.17$ | $18.89 \pm 0.22$ | $\mathbf{20.73 \pm 0.33}$ |
| **H3K79me3** | $50.76 \pm 0.16$ | $51.22 \pm 0.13$ | $\mathbf{51.99 \pm 0.18}$ | $48.80 \pm 0.14$ | $49.69 \pm 0.15$ |
| **H3K9ac** | $\mathbf{39.61 \pm 0.19}$ | $38.40 \pm 0.16$ | $39.22 \pm 0.17$ | $38.36 \pm 0.12$ | $39.30 \pm 0.25$ |
| **H4** | $67.99 \pm 0.17$ | $69.42 \pm 0.27$ | $70.62 \pm 0.26$ | $67.68 \pm 0.10$ | $\mathbf{70.75 \pm 0.21}$ |
| **H4ac** | $24.39 \pm 0.21$ | $26.74 \pm 0.31$ | $24.95 \pm 0.21$ | $24.79 \pm 0.12$ | $\mathbf{26.90 \pm 0.17}$ |

Table 5: **Compression ratio ablation: NT Histones (MCC).** Performance on histone modification prediction tasks across compression ratios $N \in \{2, 4, 8, 16, 32\}$. Values represent mean $\pm$ 95% confidence interval across cross-validation folds. Bold indicates the best result per task. Although $N = 32$ achieves the highest average MCC (39.75%) and the most per-task wins (5/10), intermediate ratios remain competitive, suggesting that aggressive compression may emphasize global chromatin patterns at the cost of local resolution.

| Task | ratio_2 | ratio_4 | ratio_8 | ratio_16 | ratio_32 |
|---|---|---|---|---|---|
| *Average* | **50.93** | 50.74 | 49.12 | 48.40 | 49.24 |
| *# Wins* | 1 | **4** | 0 | 0 | 3 |
| **enhancers** | $\mathbf{50.40 \pm 0.23}$ | $44.32 \pm 0.42$ | $45.33 \pm 0.51$ | $41.57 \pm 0.60$ | $46.10 \pm 0.37$ |
| **enhancers_types** | $32.83 \pm 0.80$ | $\mathbf{32.91 \pm 0.50}$ | $28.93 \pm 0.72$ | $27.67 \pm 0.57$ | $28.84 \pm 0.64$ |
| **promoter_all** | $80.96 \pm 0.07$ | $\mathbf{83.77 \pm 0.03}$ | $80.52 \pm 0.06$ | $79.80 \pm 0.10$ | $78.35 \pm 0.07$ |
| **promoter_no_tata** | $81.01 \pm 0.06$ | $\mathbf{83.82 \pm 0.08}$ | $81.23 \pm 0.10$ | $80.14 \pm 0.07$ | $79.14 \pm 0.11$ |
| **promoter_tata** | $78.48 \pm 0.22$ | $\mathbf{78.86 \pm 0.32}$ | $75.47 \pm 0.23$ | $75.00 \pm 0.34$ | $72.88 \pm 0.24$ |
| **splice_sites_acceptors** | $31.08 \pm 0.27$ | $31.96 \pm 0.26$ | $30.07 \pm 0.19$ | $31.30 \pm 0.22$ | $\mathbf{34.01 \pm 0.20}$ |
| **splice_sites_all** | $22.64 \pm 0.23$ | $20.97 \pm 0.25$ | $21.75 \pm 0.28$ | $23.05 \pm 0.15$ | $\mathbf{24.03 \pm 0.15}$ |
| **splice_sites_donors** | $30.01 \pm 0.16$ | $29.36 \pm 0.32$ | $29.65 \pm 0.25$ | $28.69 \pm 0.15$ | $\mathbf{30.55 \pm 0.25}$ |

Table 6: **Compression ratio ablation: NT Regulatory (MCC).** Performance on regulatory element classification tasks (enhancers, promoters, and splice sites) across compression ratios. Lower compression ($N = 2$) achieves the highest average MCC (50.93%), while $N = 4$ obtains the most per-task wins (4/8), particularly dominating promoter recognition. This pattern indicates that fine-grained local sequence information is critical for accurate regulatory element discrimination.

Figure 3 presents training dynamics at $N = 4$. The ratio loss component exhibits expected behavior: it decreases throughout training for $\alpha > 0$, indicating successful regularization toward the target compression. Notably, even at $\alpha = 0.03$, the model effectively learns the target ratio while maintaining low MLM loss.

A critical finding emerges from the $N = 8$, $\alpha = 0.0$ condition (Figure 4): without ratio loss supervision, the model's effective compression ratio diverges during training, with the ratio loss exhibiting unstable behavior. This confirms that the ratio loss is essential for maintaining the target compression – without explicit regularization, the model gravitates toward an empirically preferred compression of approximately $N \approx 4$, suggesting this ratio represents a natural equilibrium between compression and reconstruction for genomic sequences at this model scale.

Downstream performance (Tables 8–10) shows $\alpha = 0.1$ achieving the highest average on NT Histones (40.36%) and NT Regulatory (51.19%), while $\alpha = 0.03$ leads on Genomic Benchmarks (52.84%). We adopt $\alpha = 0.03$ for main experiments as it provides stable training dynamics with strong cross-benchmark performance, avoiding potential over-regularization at higher $\alpha$ values.

| Task | ratio_2 | ratio_4 | ratio_8 | ratio_16 | ratio_32 |
|------|---------|---------|---------|----------|----------|
| *Average* | 51.73 | **52.07** | 51.41 | 50.49 | 50.14 |
| *# Wins* | **4** | 3 | 1 | 0 | 1 |
| **demo_coding_vs_intergenomic_seqs** | 74.82 ± 0.04 | **75.29 ± 0.12** | 74.37 ± 0.07 | 73.82 ± 0.10 | 71.64 ± 0.07 |
| **demo_human_or_worm** | 79.28 ± 0.08 | **85.34 ± 0.05** | 80.84 ± 0.20 | 77.38 ± 0.04 | 76.69 ± 0.05 |
| **drosophila_enhancers_stark** | 35.87 ± 0.81 | 36.54 ± 0.93 | 38.44 ± 0.34 | 38.75 ± 0.54 | **38.91 ± 0.92** |
| **dummy_mouse_enhancers_ensembl** | **56.97 ± 2.61** | 56.55 ± 2.53 | 54.60 ± 1.99 | 53.74 ± 0.50 | 55.38 ± 1.57 |
| **human_enhancers_cohn** | **45.98 ± 0.27** | 44.61 ± 0.23 | 44.91 ± 0.31 | 43.60 ± 0.40 | 43.58 ± 0.17 |
| **human_enhancers_ensembl** | **43.29 ± 0.07** | 41.57 ± 0.09 | 41.95 ± 0.05 | 41.28 ± 0.15 | 40.90 ± 0.08 |
| **human_ensembl_regulatory** | **36.78 ± 0.10** | 35.30 ± 0.07 | 34.11 ± 0.04 | 34.46 ± 0.12 | 34.13 ± 0.05 |
| **human_nontata_promoters** | 65.70 ± 0.09 | **66.31 ± 0.25** | 66.25 ± 0.09 | 65.71 ± 0.19 | 63.49 ± 0.24 |
| **human_ocr_ensembl** | 26.85 ± 0.03 | 27.08 ± 0.07 | **27.23 ± 0.19** | 25.61 ± 0.10 | 26.54 ± 0.09 |

Table 7: **Compression ratio ablation: Genomic Benchmarks (MCC).** Performance across diverse genomic classification tasks under varying compression ratios. $N = 4$ achieves the highest average MCC (52.07%) with strong performance on species discrimination and promoter tasks, while $N = 2$ wins on more individual tasks (4/9). The consistent degradation at $N \geq 16$ confirms that excessive compression impairs the model's ability to capture task-relevant sequence features.

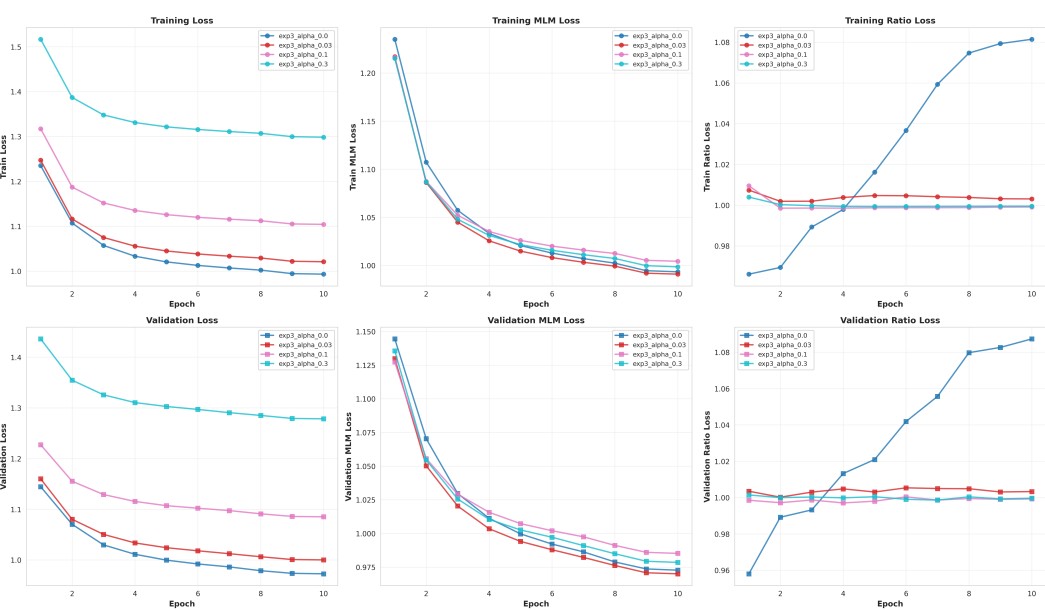

Figure 3: Training dynamics across ratio loss weights $\alpha$ at compression ratio $N = 4$. The ratio loss component (right) confirms effective regularization for $\alpha > 0$.

### B.3 CONTEXT LENGTH

Genomic regulatory elements operate across multiple length scales, from local motifs spanning tens of nucleotides to distal regulatory interactions exceeding kilobases. We evaluated context lengths $L \in \{1024, 2048, 4096, 8192\}$ byte pairs to determine the optimal receptive field for our architecture.

Training dynamics (Figure 5) reveal minimal loss differences across context lengths, with longer contexts providing marginal improvements. Downstream evaluation (Tables 11–13) presents a heterogeneous pattern: $L = 8192$ achieves the highest average on NT Histones (38.60%) and GB (53.30%), while $L = 4096$ leads on NT Regulatory tasks (49.96%).

The absence of consistent performance gains at $L = 8192$ despite increased computational cost suggests that, at the 2M parameter scale, the model's representational capacity may be the limiting factor rather than context length. We select $L = 4096$ for main experiments as it provides a favorable

| Task | alpha_0.0 | alpha_0.0_ratio_8 | alpha_0.03 | alpha_0.03_ratio_8 | alpha_0.1 | alpha_0.3 |
|---|---|---|---|---|---|---|
| *Average* | 39.23 | 38.71 | 38.46 | 38.26 | **40.36** | 39.05 |
| *# Wins* | 2 | 0 | 0 | 0 | **8** | 0 |
| **H3** | 66.01 ± 0.17 | 67.85 ± 0.16 | 66.66 ± 0.12 | 66.32 ± 0.18 | **68.13 ± 0.16** | 67.25 ± 0.25 |
| **H3K14ac** | 28.66 ± 0.24 | 27.38 ± 0.16 | 25.61 ± 0.21 | 24.95 ± 0.35 | **30.63 ± 0.32** | 27.13 ± 0.16 |
| **H3K36me3** | 38.15 ± 0.21 | 37.66 ± 0.17 | 36.94 ± 0.20 | 38.20 ± 0.17 | **39.43 ± 0.30** | 37.83 ± 0.15 |
| **H3K4me1** | **31.12 ± 0.35** | 29.98 ± 0.21 | 30.43 ± 0.15 | 28.91 ± 0.23 | 29.71 ± 0.30 | 30.79 ± 0.28 |
| **H3K4me2** | 23.44 ± 0.24 | 24.71 ± 0.15 | 24.92 ± 0.23 | 24.46 ± 0.21 | **25.09 ± 0.22** | 24.25 ± 0.23 |
| **H3K4me3** | 19.60 ± 0.15 | 18.40 ± 0.29 | 18.82 ± 0.25 | 18.59 ± 0.15 | **20.17 ± 0.25** | 18.74 ± 0.29 |
| **H3K79me3** | 50.92 ± 0.15 | 48.70 ± 0.14 | 50.72 ± 0.13 | 51.14 ± 0.14 | **52.33 ± 0.18** | 50.39 ± 0.14 |
| **H3K9ac** | **39.56 ± 0.18** | 38.90 ± 0.10 | 36.35 ± 0.20 | 37.80 ± 0.12 | 39.47 ± 0.16 | 38.99 ± 0.24 |
| **H4** | 69.44 ± 0.18 | 69.80 ± 0.19 | 70.37 ± 0.19 | 67.39 ± 0.31 | **71.34 ± 0.12** | 70.27 ± 0.24 |
| **H4ac** | 25.43 ± 0.15 | 23.76 ± 0.09 | 23.74 ± 0.13 | 24.85 ± 0.26 | **27.31 ± 0.17** | 24.86 ± 0.24 |

Table 8: **Ratio Loss Weight Ablation: NT Histones (MCC).** Comparison across $\alpha$ values at $N = 4$ and $N = 8$. The $\alpha = 0.1$ configuration achieves strongest average performance on histone modification prediction.

| Task | alpha_0.0 | alpha_0.0_ratio_8 | alpha_0.03 | alpha_0.03_ratio_8 | alpha_0.1 | alpha_0.3 |
|---|---|---|---|---|---|---|
| *Average* | 50.36 | 49.16 | 50.83 | 50.47 | **51.19** | 50.53 |
| *# Wins* | 0 | 0 | 2 | 2 | **3** | 1 |
| **enhancers** | 48.11 ± 0.50 | 46.41 ± 0.37 | 45.73 ± 0.38 | **48.87 ± 0.56** | 46.01 ± 0.45 | 47.23 ± 0.35 |
| **enhancers_types** | 30.32 ± 0.38 | 29.17 ± 0.45 | **33.15 ± 0.32** | 30.90 ± 0.43 | 29.81 ± 0.52 | 26.98 ± 0.37 |
| **promoter_all** | 81.92 ± 0.05 | 79.86 ± 0.08 | 81.34 ± 0.08 | 81.07 ± 0.05 | **83.39 ± 0.07** | 82.65 ± 0.10 |
| **promoter_no_tata** | 82.15 ± 0.09 | 80.05 ± 0.08 | 81.41 ± 0.05 | 81.28 ± 0.07 | **83.57 ± 0.06** | 82.91 ± 0.09 |
| **promoter_tata** | 77.48 ± 0.33 | 77.17 ± 0.31 | 76.59 ± 0.34 | 74.30 ± 0.25 | **82.35 ± 0.33** | 79.17 ± 0.22 |
| **splice_sites_acceptors** | 29.91 ± 0.18 | 31.25 ± 0.21 | 33.20 ± 0.32 | **33.88 ± 0.19** | 31.74 ± 0.23 | 33.17 ± 0.36 |
| **splice_sites_all** | 22.31 ± 0.21 | 20.96 ± 0.26 | **24.88 ± 0.16** | 22.36 ± 0.18 | 22.11 ± 0.25 | 19.98 ± 0.23 |
| **splice_sites_donors** | 30.72 ± 0.27 | 28.43 ± 0.33 | 30.30 ± 0.30 | 31.10 ± 0.32 | 30.55 ± 0.27 | **32.12 ± 0.28** |

Table 9: **Ratio Loss Weight Ablation: NT Regulatory (MCC).** Moderate ratio loss weights ($\alpha \in \{0.03, 0.1\}$) consistently outperform extreme values on regulatory element tasks.

| Task | alpha_0.0 | alpha_0.0_ratio_8 | alpha_0.03 | alpha_0.03_ratio_8 | alpha_0.1 | alpha_0.3 |
|---|---|---|---|---|---|---|
| *Average* | 51.71 | 51.93 | **52.84** | 51.62 | 51.76 | 52.44 |
| *# Wins* | 1 | 1 | **3** | 1 | 1 | 2 |
| **demo_coding_vs_intergenomic_seqs** | **75.43 ± 0.13** | 74.96 ± 0.07 | 74.72 ± 0.06 | 74.14 ± 0.06 | 74.98 ± 0.10 | 74.22 ± 0.07 |
| **demo_human_or_worm** | 83.27 ± 0.08 | 79.44 ± 0.06 | **83.83 ± 0.08** | 80.50 ± 0.07 | 78.47 ± 0.04 | 81.18 ± 0.12 |
| **drosophila_enhancers_stark** | 39.64 ± 0.56 | 39.50 ± 0.66 | **41.01 ± 0.79** | 39.36 ± 0.52 | 38.69 ± 0.42 | 37.80 ± 0.78 |
| **dummy_mouse_enhancers_ensembl** | 55.13 ± 1.49 | 58.24 ± 2.05 | 59.69 ± 2.92 | 54.89 ± 3.59 | 59.00 ± 2.30 | **62.29 ± 1.67** |
| **human_enhancers_cohn** | 44.53 ± 0.35 | **45.80 ± 0.19** | 45.76 ± 0.20 | 45.60 ± 0.17 | 45.32 ± 0.19 | 45.29 ± 0.27 |
| **human_enhancers_ensembl** | 41.82 ± 0.09 | 42.41 ± 0.09 | **43.01 ± 0.09** | 41.81 ± 0.08 | 42.14 ± 0.09 | 42.83 ± 0.12 |
| **human_ensembl_regulatory** | 34.05 ± 0.03 | 34.31 ± 0.08 | 35.20 ± 0.06 | 35.16 ± 0.03 | 34.85 ± 0.04 | **35.74 ± 0.07** |
| **human_nontata_promoters** | 65.35 ± 0.20 | 65.98 ± 0.23 | 66.19 ± 0.10 | **66.69 ± 0.32** | 65.45 ± 0.06 | 65.95 ± 0.20 |
| **human_ocr_ensembl** | 26.19 ± 0.09 | 26.70 ± 0.14 | 26.16 ± 0.12 | 26.42 ± 0.15 | **26.95 ± 0.15** | 26.66 ± 0.09 |

Table 10: **Ratio Loss Weight Ablation: Genomic Benchmarks (MCC).** $\alpha = 0.03$ achieves the highest average (52.84%), representing the configuration used in main experiments.

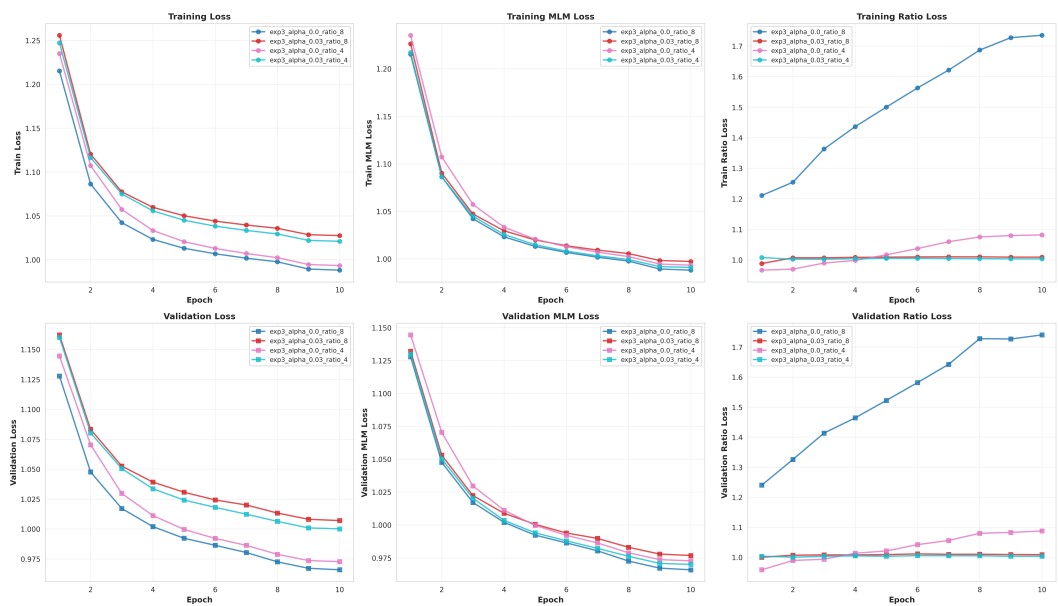

Figure 4: Training dynamics at $N = 8$ comparing $\alpha = 0.0$ versus $\alpha = 0.03$. Without ratio loss ($\alpha = 0.0$), the ratio loss diverges, indicating the model fails to maintain the target compression ratio.

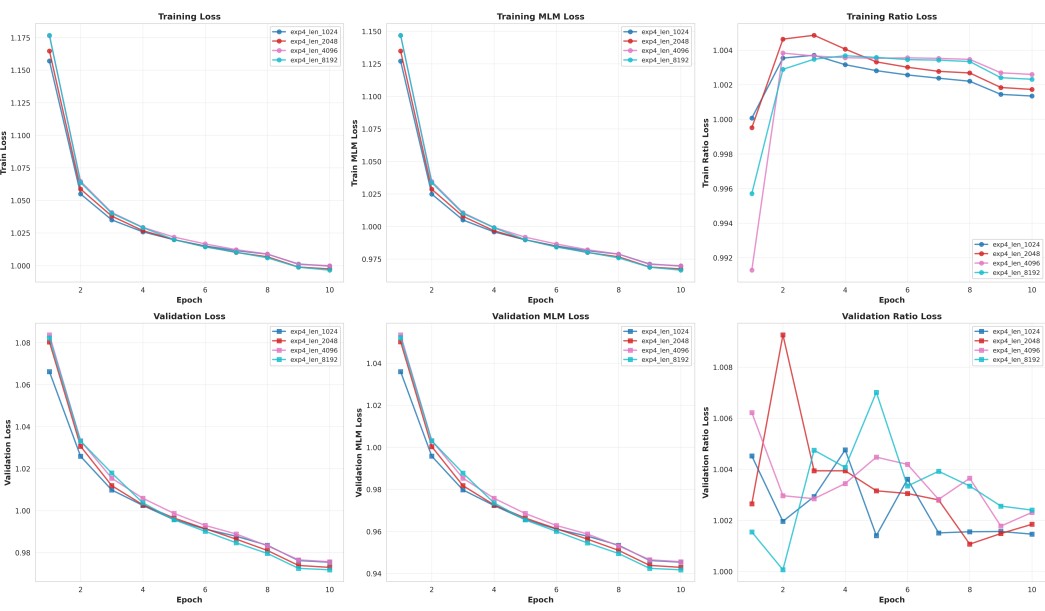

Figure 5: Training loss across context lengths $L \in \{1024, 2048, 4096, 8192\}$. Longer contexts yield marginally lower loss, though differences diminish beyond $L = 4096$.

trade-off between computational efficiency and performance, achieving the best results on regulatory element classification while remaining competitive across all benchmarks.

### B.4 ARCHITECTURE

We compare three architectural variants to assess the contribution of each component: (1) **Hybrid**: interleaved Mamba-2 and Transformer layers, (2) **Pure Mamba**: exclusively BiMamba-2 layers,

| Task | context_1024 | context_2048 | context_4096 | context_8192 |
|---|---|---|---|---|
| *Average* | 38.29 | 37.28 | 38.03 | **38.60** |
| *# Wins* | 3 | 2 | 1 | **4** |
| **H3** | $65.63 \pm 0.21$ | **$67.38 \pm 0.16$** | $67.18 \pm 0.23$ | $65.97 \pm 0.13$ |
| **H3K14ac** | $25.11 \pm 0.18$ | $24.22 \pm 0.21$ | $25.84 \pm 0.26$ | **$26.28 \pm 0.21$** |
| **H3K36me3** | $35.70 \pm 0.25$ | $34.56 \pm 0.11$ | $35.97 \pm 0.13$ | **$37.50 \pm 0.25$** |
| **H3K4me1** | $28.23 \pm 0.22$ | $28.26 \pm 0.32$ | **$29.30 \pm 0.15$** | $28.92 \pm 0.14$ |
| **H3K4me2** | $24.59 \pm 0.22$ | $24.53 \pm 0.23$ | $24.85 \pm 0.30$ | **$26.01 \pm 0.21$** |
| **H3K4me3** | $19.88 \pm 0.34$ | $16.11 \pm 0.27$ | $19.71 \pm 0.46$ | **$20.39 \pm 0.27$** |
| **H3K79me3** | **$50.54 \pm 0.17$** | $48.68 \pm 0.18$ | $49.37 \pm 0.12$ | $49.79 \pm 0.15$ |
| **H3K9ac** | $38.36 \pm 0.19$ | **$38.42 \pm 0.15$** | $37.70 \pm 0.24$ | $37.75 \pm 0.14$ |
| **H4** | **$68.43 \pm 0.16$** | $67.42 \pm 0.17$ | $65.90 \pm 0.16$ | $67.49 \pm 0.18$ |
| **H4ac** | **$26.48 \pm 0.16$** | $23.22 \pm 0.28$ | $24.53 \pm 0.21$ | $25.94 \pm 0.31$ |

Table 11: **Context Length Ablation: NT Histones (MCC).** Extended context ($L = 8192$) provides modest improvements on histone tasks, potentially capturing longer-range chromatin dependencies.

| Task | context_1024 | context_2048 | context_4096 | context_8192 |
|---|---|---|---|---|
| *Average* | 47.76 | 49.57 | **49.96** | 48.38 |
| *# Wins* | 0 | **4** | 2 | 2 |
| **enhancers** | $45.53 \pm 0.29$ | $46.01 \pm 0.34$ | $45.40 \pm 0.28$ | **$47.11 \pm 0.37$** |
| **enhancers_types** | $28.21 \pm 0.35$ | $30.35 \pm 0.61$ | $30.47 \pm 0.32$ | **$30.78 \pm 0.44$** |
| **promoter_all** | $79.26 \pm 0.06$ | **$81.29 \pm 0.06$** | $81.00 \pm 0.08$ | $78.68 \pm 0.06$ |
| **promoter_no_tata** | $79.73 \pm 0.09$ | **$81.64 \pm 0.10$** | $81.31 \pm 0.08$ | $79.04 \pm 0.08$ |
| **promoter_tata** | $71.45 \pm 0.30$ | $73.31 \pm 0.33$ | **$76.31 \pm 0.20$** | $73.04 \pm 0.19$ |
| **splice_sites_acceptors** | $29.47 \pm 0.29$ | **$32.36 \pm 0.21$** | $31.51 \pm 0.38$ | $30.88 \pm 0.23$ |
| **splice_sites_all** | $20.01 \pm 0.19$ | $21.68 \pm 0.23$ | **$24.12 \pm 0.29$** | $20.66 \pm 0.21$ |
| **splice_sites_donors** | $28.43 \pm 0.22$ | **$29.93 \pm 0.29$** | $29.56 \pm 0.26$ | $26.84 \pm 0.24$ |

Table 12: **Context Length Ablation: NT Regulatory (MCC).** $L = 4096$ achieves optimal performance on regulatory element tasks, suggesting this length captures the relevant sequence context for promoter and splice site recognition.

and (3) **Pure Transformer**: exclusively attention layers. All configurations maintain approximately 2M parameters through adjusted layer counts.

The architectural configurations are defined as follows:

- **Hybrid**: Two-stage architecture with layout `[m2t1, [M8], m3]`, combining 2 Mamba layers and 1 Transformer layer in the first stage, 8 Mamba layers in the main backbone, and 3 Mamba layers in the final stage. Model dimensions are $d_{\text{model}} = [64, 128]$ with intermediate FFN dimension 384 in the main backbone.

- **Pure Mamba**: Layout `[m3, [M8], m3]` replacing all Transformer layers with BiMamba-2 blocks while maintaining identical dimensionality ($d_{\text{model}} = [64, 128]$, $d_{\text{intermediate}} = 384$).

- **Pure Transformer**: Layout `[t3, [T8], t3]` substituting all Mamba layers with multi-head attention blocks. Attention configurations use $[2, 4]$ heads with rotary embeddings of dimension $[8, 16]$.

Results (Tables 14–16) reveal architecture-dependent performance patterns. The hybrid model achieves the best average on NT Histones (38.66%) and GB (52.82%), while pure Mamba excels on NT Regulatory tasks (51.12%). Pure Transformer consistently underperforms, with average MCC reductions of 2.95%, 4.51%, and 5.10% on NT Histones, NT Regulatory, and GB respectively, compared to the hybrid architecture.

| Task | context_1024 | context_2048 | context_4096 | context_8192 |
|---|---|---|---|---|
| *Average* | 51.78 | 52.53 | 51.39 | **53.30** |
| *# Wins* | 1 | 3 | 1 | **4** |
| **demo_coding_vs_intergenomic_seqs** | **75.69 ± 0.02** | 75.46 ± 0.05 | 74.83 ± 0.05 | 75.39 ± 0.06 |
| **demo_human_or_worm** | 83.75 ± 0.06 | 82.45 ± 0.14 | 80.74 ± 0.11 | **83.76 ± 0.11** |
| **drosophila_enhancers_stark** | 39.14 ± 0.24 | **41.07 ± 0.63** | 38.05 ± 0.33 | 38.81 ± 0.47 |
| **dummy_mouse_enhancers_ensembl** | 50.14 ± 1.33 | 55.20 ± 0.83 | 50.54 ± 0.75 | **60.39 ± 1.74** |
| **human_enhancers_cohn** | 45.34 ± 0.30 | **46.22 ± 0.22** | 45.79 ± 0.13 | 45.95 ± 0.12 |
| **human_enhancers_ensembl** | 42.68 ± 0.08 | 42.85 ± 0.13 | **43.37 ± 0.06** | 42.92 ± 0.06 |
| **human_ensembl_regulatory** | 35.68 ± 0.06 | 34.94 ± 0.06 | 35.42 ± 0.07 | **37.14 ± 0.08** |
| **human_nontata_promoters** | 65.52 ± 0.22 | **67.10 ± 0.18** | 65.52 ± 0.09 | 66.49 ± 0.19 |
| **human_ocr_ensembl** | 28.09 ± 0.19 | 27.41 ± 0.06 | 28.28 ± 0.07 | **28.82 ± 0.14** |

Table 13: **Context Length Ablation: Genomic Benchmarks (MCC).** Longer contexts generally improve performance, with $L = 8192$ achieving the highest average (53.30%).

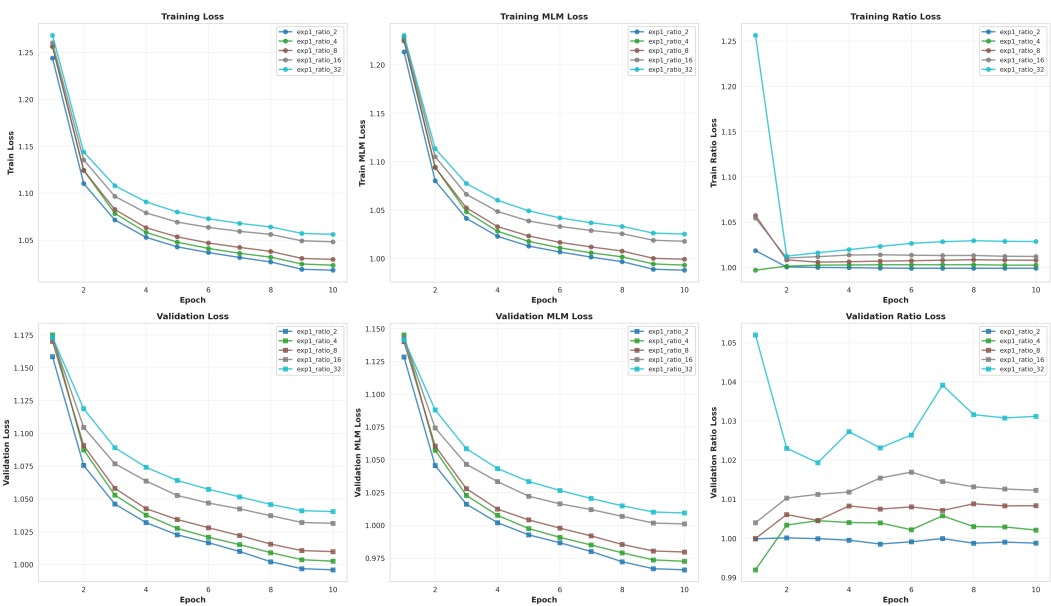

Figure 6: Training loss comparison across architectural variants. The hybrid architecture achieves competitive loss while combining the strengths of both SSM and attention mechanisms.

The strong performance of pure Mamba on regulatory element tasks – particularly enhancers (49.61% vs. 43.26% for hybrid) and promoters (81.72% vs. 79.95%) – suggests that SSM's efficient long-range modeling is particularly beneficial for these tasks. However, the hybrid architecture's superior performance on tasks requiring precise local pattern recognition (e.g., splice site donors: 32.24% vs. 29.28%) indicates that attention mechanisms contribute complementary capabilities.

We adopt the hybrid architecture for main experiments following prior work Hwang et al. (2025a) demonstrating the efficacy of Mamba-Transformer hybrids for sequence modeling. However, these results indicate that pure Mamba architectures represent a promising direction for future investigation, potentially offering improved performance with reduced computational complexity for genomic applications.

| Task | hybrid | pure_mamba | pure_transformer |
|---|---|---|---|
| *Average* | **38.66** | 38.23 | 35.71 |
| *# Wins* | **5** | 4 | 1 |
| **H3** | **68.47 ± 0.17** | 67.11 ± 0.17 | 66.59 ± 0.19 |
| **H3K14ac** | **27.07 ± 0.13** | 25.42 ± 0.19 | 23.17 ± 0.18 |
| **H3K36me3** | 37.52 ± 0.13 | **38.27 ± 0.20** | 31.92 ± 0.32 |
| **H3K4me1** | **30.05 ± 0.11** | 27.45 ± 0.17 | 27.70 ± 0.31 |
| **H3K4me2** | 24.26 ± 0.18 | **24.92 ± 0.13** | 22.70 ± 0.22 |
| **H3K4me3** | 17.88 ± 0.27 | **18.49 ± 0.18** | 17.65 ± 0.18 |
| **H3K79me3** | **50.52 ± 0.20** | 49.83 ± 0.16 | 42.23 ± 0.18 |
| **H3K9ac** | 37.22 ± 0.08 | 37.63 ± 0.14 | **38.64 ± 0.22** |
| **H4** | **68.01 ± 0.09** | 66.42 ± 0.16 | 63.16 ± 0.17 |
| **H4ac** | 25.61 ± 0.14 | **26.75 ± 0.14** | 23.31 ± 0.35 |

Table 14: **Architecture Ablation: NT Histones (MCC).** The hybrid architecture achieves the highest average (38.66%) with balanced performance across histone modification tasks.

| Task | hybrid | pure_mamba | pure_transformer |
|---|---|---|---|
| *Average* | 48.48 | **51.12** | 46.61 |
| *# Wins* | 2 | **6** | 0 |
| **enhancers** | 43.26 ± 0.42 | **49.61 ± 0.26** | 47.75 ± 0.46 |
| **enhancers_types** | 25.26 ± 0.35 | **33.18 ± 0.52** | 31.21 ± 0.60 |
| **promoter_all** | 79.95 ± 0.09 | **81.72 ± 0.06** | 75.94 ± 0.05 |
| **promoter_no_tata** | 80.08 ± 0.13 | **82.27 ± 0.11** | 76.92 ± 0.12 |
| **promoter_tata** | 74.95 ± 0.31 | **77.95 ± 0.37** | 66.42 ± 0.15 |
| **splice_sites_acceptors** | **30.88 ± 0.19** | 30.68 ± 0.30 | 26.57 ± 0.21 |
| **splice_sites_all** | 21.20 ± 0.17 | **24.29 ± 0.16** | 20.65 ± 0.21 |
| **splice_sites_donors** | **32.24 ± 0.23** | 29.28 ± 0.24 | 27.43 ± 0.28 |

Table 15: **Architecture Ablation: NT Regulatory (MCC).** Pure Mamba achieves substantially higher performance (51.12%) on regulatory element classification, suggesting SSM's long-range modeling is particularly effective for these tasks.

| Task | hybrid | pure_mamba | pure_transformer |
|---|---|---|---|
| *Average* | **52.82** | 52.46 | 47.72 |
| *# Wins* | **6** | 3 | 0 |
| **demo_coding_vs_intergenomic_seqs** | **75.14 ± 0.07** | 74.57 ± 0.09 | 68.45 ± 0.03 |
| **demo_human_or_worm** | **84.62 ± 0.11** | 80.35 ± 0.11 | 63.37 ± 0.14 |
| **drosophila_enhancers_stark** | **40.12 ± 0.69** | 40.06 ± 0.31 | 37.31 ± 0.30 |
| **dummy_mouse_enhancers_ensembl** | 57.57 ± 2.89 | **61.09 ± 2.32** | 55.03 ± 2.68 |
| **human_enhancers_cohn** | **46.09 ± 0.05** | 45.50 ± 0.19 | 43.61 ± 0.33 |
| **human_enhancers_ensembl** | 42.08 ± 0.11 | **42.50 ± 0.05** | 40.74 ± 0.10 |
| **human_ensembl_regulatory** | **35.55 ± 0.04** | 34.57 ± 0.04 | 32.04 ± 0.06 |
| **human_nontata_promoters** | 66.20 ± 0.19 | **66.21 ± 0.14** | 62.86 ± 0.21 |
| **human_ocr_ensembl** | **28.03 ± 0.06** | 27.25 ± 0.09 | 26.11 ± 0.16 |

Table 16: **Architecture Ablation: Genomic Benchmarks (MCC).** The hybrid architecture (52.82%) marginally outperforms pure Mamba (52.46%), while pure Transformer lags substantially (47.72%).

